# The African swine fever virus p22 inhibits the JAK-STAT signaling pathway by promoting the TAX1BP1-mediated degradation of the type I interferon receptor

Haojie Ren[1ʘ], Yanjin Wang[2ʘ], Lian-Feng Li[2ʘ], Lan-Fang Shi[1], Yu-He Ma[1], Jun-Hao Fan[1], Xiao-Ya Pan[1], Han-Cheng Shao[1], Yuhang Zhang[1,3,4], Shichong Han[1,3,4], Bo Wan[1,3,4], Hua-Ji Qiu[2]*, Gai-Ping Zhang[1,3]*, Su Li[2]*, Wen-Rui He [1,3,4]*

1 International Joint Research Centre of National Animal Immunology, College of Veterinary Medicine, Henan Agricultural University, Zhengzhou, Henan, P.R. China, 2 State Key Laboratory for Animal Disease Control and Prevention, National African Swine Fever Para-Reference Laboratory, National High-Containment Facilities for Animal Disease Control and Prevention, Harbin Veterinary Research Institute, Chinese Academy of Agricultural Sciences, Harbin, Heilongjiang, P.R. China, 3 Longhu Laboratory, Zhengzhou, Henan, P.R. China, 4 Ministry of Education Key Laboratory for Animal Pathogens and Biosafety, Henan Agricultural University, Zhengzhou, P.R. China

ʘ These authors contributed equally to this work.
* wrhe0111@163.com (WRH); lisu@caas.cn (SL); zhanggaiping2003@163.com (GPZ); qiuhuaji@caas.cn (HJQ)

## Abstract

African swine fever virus (ASFV) is the causative agent of African swine fever (ASF), a devastating disease epidemic in Asia and Europe. Large knowledge gaps regarding the biological characteristics of viral structural proteins have severely hindered the development of vaccines against ASF. The p22 protein, an internal envelope membrane protein of ASFV, is one such protein that is yet to be deciphered despite its significance. Here, our results indicated that p22 is not essential for the morphogenesis and replication of ASFV in porcine alveolar macrophages. The ASFV p22 negatively regulates the IFN-$\beta$-triggered activation of the Janus kinase-signal transducer and activator of transcription (JAK-STAT) signaling pathway. Mechanistically, the ASFV p22 promotes the association of the Tax1-binding protein 1 (TAX1BP1) with the type I IFN receptor 1 (IFNAR1) *via* its transmembrane region, thereby facilitating the autophagic degradation of IFNAR1 and impairing the host antiviral responses at the initial step of JAK-STAT signaling pathway. These findings clarify the biological functions of p22 in ASFV replication and uncover a novel autophagy degradation mechanism for IFNAR1, which provide a novel theoretical basis for understanding the biological characteristics of ASFV and may contribute to the development of vaccines and antiviral therapies against ASF.

**Data availability statement:** The RNA sequence reads generated in this study have been deposited in the sequence read archive database under the accession number PRJNA1225612 (BioProject). The complete genome sequence of the ASFV HLJ/18 strain used in this study can be accessed under the GenBank accession no. MK333180.1.

**Funding:** This work was supported by the National Natural Science Foundation of China (grant 32102655, WRH) and the Major Scientific and Technological Project of Henan Province, China (grant 221100110600, GPZ). The funders had no role in study design, data collection and analysis, decision to publish, or preparation of the manuscript. None of the authors received a salary from the funders.

**Competing interests:** The authors have declared that no competing interests exist.

## Author summary

ASFV is a highly pathogenic agent, with a mortality rate of up to 100% for acute infection, which has led to significant economic losses in the global swine industry. However, large knowledge gaps regarding the composition of infectious virions and the biological characteristics of viral proteins currently act as major stumbling blocks to the development of vaccines against this devastating disease. The p22 protein, an internal envelope membrane protein of ASFV, is one such protein that is yet to be deciphered despite its significance. Here, we discovered that the selective autophagy receptor TAX1BP1 interacted with IFNAR1 and facilitated its degradation through the autophagy pathway. The transmembrane region of the ASFV p22 enhances the binding of TAX1BP1 to IFNAR1, thereby facilitating the degradation of IFNAR1. This further led to a decrease in the phosphorylation of key molecules in the JAK-STAT pathway and a decrease in the transcription of antiviral genes, ultimately inhibiting the host antiviral response. This study elucidates the function of p22 in regulating the IFN response after ASFV infection and provides a novel theoretical basis for understanding the biological characteristics of ASFV.

## Introduction

Autophagy is a conserved eukaryotic degradation system that sequesters intracellular components into autophagosomes, which are then fused with the lysosomes to degrade contents for recycling [1]. Although autophagy was initially considered a non-specific autodigestive response to nutrient depletion, it has become evident that autophagy also selectively recognizes and degrades 'tagged' cargos *via* autophagy receptors, such as neighbor of Brca1 (NBR1), calcium-binding and coiled-coil domain 2 (CALCOCO2/NDP52), Tax1-binding protein 1 (TAX1BP1/CALCOCO3/T6BP), optineurin (OPTN), sequestosome 1 (SQSTM1/p62), NIP3-like protein X (NIX), and Toll-interacting protein (TOLLIP) [2]. Selective autophagy is a double-edged sword in the game between viruses and hosts [3,4]. Host cells possess the ability to counteract viral invasion by degrading essential viral proteins. However, viruses can also achieve effective proliferation by hijacking the host autophagy. For instance, the host protein-sorting nexin-5 restricts the replication of the Sindbis virus, West Nile virus, and Chikungunya virus by increasing the lipid kinase activity of phosphatidylinositol-3-kinase class III complex 1 and initiating autophagosome formation [5]. However, the open reading frame 7a protein of the SARS-CoV-2 virus triggers autophagy and hinders the fusion between autophagosomes and lysosomes by degrading the synaptosome-associated protein 29, thereby promoting viral replication [6]. Picornavirus VP3 protein induces autophagy through the TP53-BAD-BAX axis to promote viral replication [7]. In addition, the influenza A virus nucleoprotein-mediated mitophagy leads to the degradation of the mitochondria-anchored protein MAVS, thereby blocking MAVS-mediated antiviral signaling and promoting virus

replication [8]. During African swine fever virus (ASFV) infection, several viral proteins, including pE199L, pMGF300-2R, p17, pMGF300-4L, pL83L, and pI215L, have been identified to participate in regulating autophagy. Specifically, the ASFV pE199L induces complete autophagy through interaction with Pyrroline-5-carboxylate reductase 2 (PYCR2) and down-regulate the expression level of PYCR2 [9]; the p17 promoted mitophagy by facilitating the interaction between TOMM70 and SQSTM1 and inhibited the innate immune response by degrading MAVS [10]; the MGF300-2R protein induced the degradation of IKKα and IKKβ through TOLLIP-mediated autophagy, which in turn negatively modulated the NF-κB signaling and promoted ASFV replication [11]; the MGF300-4L protein is linked to viral pathogenicity through its promotion of IκB kinase beta (IKKβ) autophagic degradation [12]; the L83L protein negatively regulates the cyclic GMP-AMP synthase-stimulator of interferon genes (cGAS-STING)-mediated type I interferons (IFNs) pathway by recruiting TOLLIP to facilitate STING autophagic degradation [13]. Whereas, the I215L protein inhibits type I IFN signaling by targeting interferon regulatory factor 9 (IRF9) for autophagic degradation [14]. Collectively, these viral proteins exploit various autophagy receptors (*e.g.*, p62, TOLLIP, CALCOCO2) to degrade host factors, thereby promoting viral replication. Although several studies have reported the involvement of autophagy during African swine fever virus (ASFV) infection [15], the precise relationship between ASFV and autophagy has not yet been elucidated, necessitating further investigation.

The innate immune response is the first line of host defense against viral infections [16]. Type I IFNs are recognized by the IFN receptor (IFNAR) complex, IFNAR1-IFNAR2. Then this complex activates the Janus kinase-signal transducer and activator of transcription (JAK-STAT) signaling pathway, followed by the subsequent transcription and expression of downstream antiviral genes to effectively counteract pathogen invasion [17–20]. The density of IFN receptors on the cell surface determines cellular sensitivity to IFNs and efficiency of the host antiviral response [21]. The pathogenesis of autoimmune diseases such as lupus erythematosus is closely associated with the upregulation of IFNAR1. Conversely, decreased expression of IFNAR1 results in increased tumor development and growth, as well as impairs host antiviral immunity [22]. It is known that, the phosphorylation and ubiquitination of IFNAR1 are upregulated upon viral infection, followed by endocytosis and lysosomal degradation of IFNAR1 [22,23]. Most viruses possess the ability to selectively target IFNAR1, leading to an effective reduction in host antiviral responses. Respiratory syncytial virus and feline calicivirus selectively degrade *IFNAR1* mRNA, thereby reducing its expression and blocking type I IFN-induced activation of the JAK-STAT signaling pathway [24,25]. Hepatitis B virus infection induces the production of matrix metalloproteinase 9, which in turn promotes Hepatitis B virus replication by interacting with IFNAR1 and facilitating the degradation of IFNAR1 [26]. Therefore, the maintenance of the IFN receptor stability in host cells is crucial for effective antiviral immunity. Further in-depth studies on the mechanisms involved in regulating the stability of IFN receptors are required to facilitate the development of novel diagnostic strategies for infectious diseases and autoimmune diseases.

ASFV is the only known DNA arbovirus. The ASFV genome consists of a linear double-stranded DNA molecule that is approximately 170–194 kb in length and encodes more than 160 viral proteins [27,28]. Various proteins of this enormous virus are involved in viral genome replication, transcriptional regulation, virion assembly, and immune evasion [28,29]. However, large knowledge gaps regarding the composition of infectious virions and the biological characteristics of viral proteins currently act as major stumbling blocks to the development of vaccines against this devastating disease. A membrane protein of ASFV, p22, encoded by the *KP177R* gene, is expressed early during the early stage of viral replication and is located in the internal capsule membrane of virions [30,31]. The *KP177R* gene deletion had no effect on viral replication and virulence of ASFV Georgia2010 in swine [32]. However, proteomic analysis shows that the potential p22-interacting proteins are involved in cellular metabolism, gene transcription, energy homeostasis, and autophagy, which are closely associated with viral replication [33]. Thus, further study on the biological characteristics and function of p22 is urgently needed to help unravel the sophisticated mechanisms of ASFV.

In this study, we aimed to investigate the role of p22 in viral replication and in regulating the IFN response after ASFV infection. To this end, we evaluated the inhibitory effect of p22 on the JAK-STAT pathway by analyzing the transcription of IFN-stimulated genes (*ISGs*) and phosphorylation of key molecules within this signaling cascade using the

*KP177R*-deleted ASFV mutant as well as ectopically expressed p22. This study provides a novel theoretical foundation for understanding the biological characteristics of ASFV proteins. These efforts will contribute to the development of vaccines and antiviral agents against ASF.

## Results

### The *KP177R* gene is not essential for ASFV replication in porcine alveolar macrophages (PAMs)

As an internal capsule membrane protein of ASFV, we find that p22 was present in the cytoplasm, exhibited a perinuclear distribution, and localized in the endoplasmic reticulum (ER), Golgi apparatus, and lysosomes, however, its presence was negligible in the mitochondria and proteasomes in ASFV-infected PAMs (S1 Fig).

To clarify the functional role of p22 in ASFV replication in PAMs, the *KP177R* gene was deleted from the wild-type highly virulent ASFV HLJ/2018 (ASFV-WT) strain to generate the ASFV mutant ASFV-ΔKP177R [34]. After ten rounds of limiting dilution based on the enhanced green fluorescent protein (EGFP) expression, ASFV-ΔKP177R was successfully purified (Fig 1A). Next-generation sequencing results proved the complete deletion of the *KP177R* gene in ASFV-ΔKP177R (Fig 1B). Consistent with the gene sequencing results, western blotting analysis showed that the expression of p22 was undetectable in the ASFV-ΔKP177R-infected PAMs (Fig 1C), indicating that ASFV-ΔKP177R was successfully generated. ASFV-ΔKP177R and ASFV-WT were then posed to transmission electron microscopy and immunoelectron microscopy, and the results showed that p22 is indeed an inner envelope membrane protein of ASFV (red arrow) (Fig 1D), and that its deletion had no effect on virion morphology (red arrow) (Fig 1E). The "rosettes" formation of red blood cells on was observed on the surface of the ASFV-ΔKP177R- or ASFV-WT-infected PAMs, and no significant difference was detected in the number of rosettes between the two groups, indicating that the deletion of the *KP177R* gene does not affect the hemadsorption property of ASFV (Fig 1F). Furthermore, the growth kinetics of ASFV-ΔKP177R were similar to those of ASFV-WT depending on the time point considered (Fig 1G), indicating that the *KP177R* gene deletion does not affect the replication of ASFV in PAMs. These results suggest that the *KP177R* gene is not essential for ASFV morphogenesis and replication in PAMs.

### The *KP177R*-deleted ASFV mutant significantly promotes activation of the JAK-STAT signaling pathway in PAMs

To further characterize the biological function of p22, the differentially expressed genes (DEGs) between PAMs infected with ASFV-ΔKP177R and those infected with ASFV-WT at a multiplicity of infection (MOI) of 3 were systemically analyzed *via* RNA sequencing (RNA-seq) at 12 and 18 hours post infection (hpi) (The data are available in the Sequence Read Archive under the accession number PRJNA1225612). To analyze the pathways associated with p22, Kyoto Encyclopedia of Genes and Genomes (KEGG) enrichment analyses were performed. Compared with that in the ASFV-WT-infected PAMs, the upregulated genes in the ASFV-ΔKP177R-infected PAMs were most enriched several signaling pathways including the RAS signaling, the MAPK signaling, and JAK-STAT signaling cascade (red rectangle) (Fig 2A). To verify the credibility of our RNA-seq data, six DEGs of the top enriched pathways were selected for examination of mRNA transcription using RT-qPCR assay. Specifically, the selected genes included SHC adaptor protein 3 (*SHC3*) and RAS protein activator like 1 (*RASAL1*) from the RAS signaling pathway, calcium voltage-gated channel subunit alpha1 A (*CACNA1A*) and *CACNA1C* from the MAPK signaling pathway, as well as *ISG15* and *ISG20* from the JAK-STAT signaling pathway. Our results suggested that the transcriptional levels of these genes were significantly upregulated in the ASFV-ΔKP177R infection group compared with the ASFV-WT infection group, thereby corroborating the reliability of the RNA-seq data (S2A Fig). Considering the intimate connection between the pathogenicity of ASFV and its ability to counteract host IFN responses [35], we focused on and evaluated the transcript levels of *ISGs* downstream of the JAK-STAT signaling pathway. Remarkably, compared with those in the ASFV-WT-infected PAMs, the transcriptional levels of *ISGs* were notably up-regulated in the ASFV-ΔKP177R-infected PAMs at 12 and 18 hpi, respectively (Fig 2B). These results indicate that p22 is involved in the JAK-STAT signaling pathway.

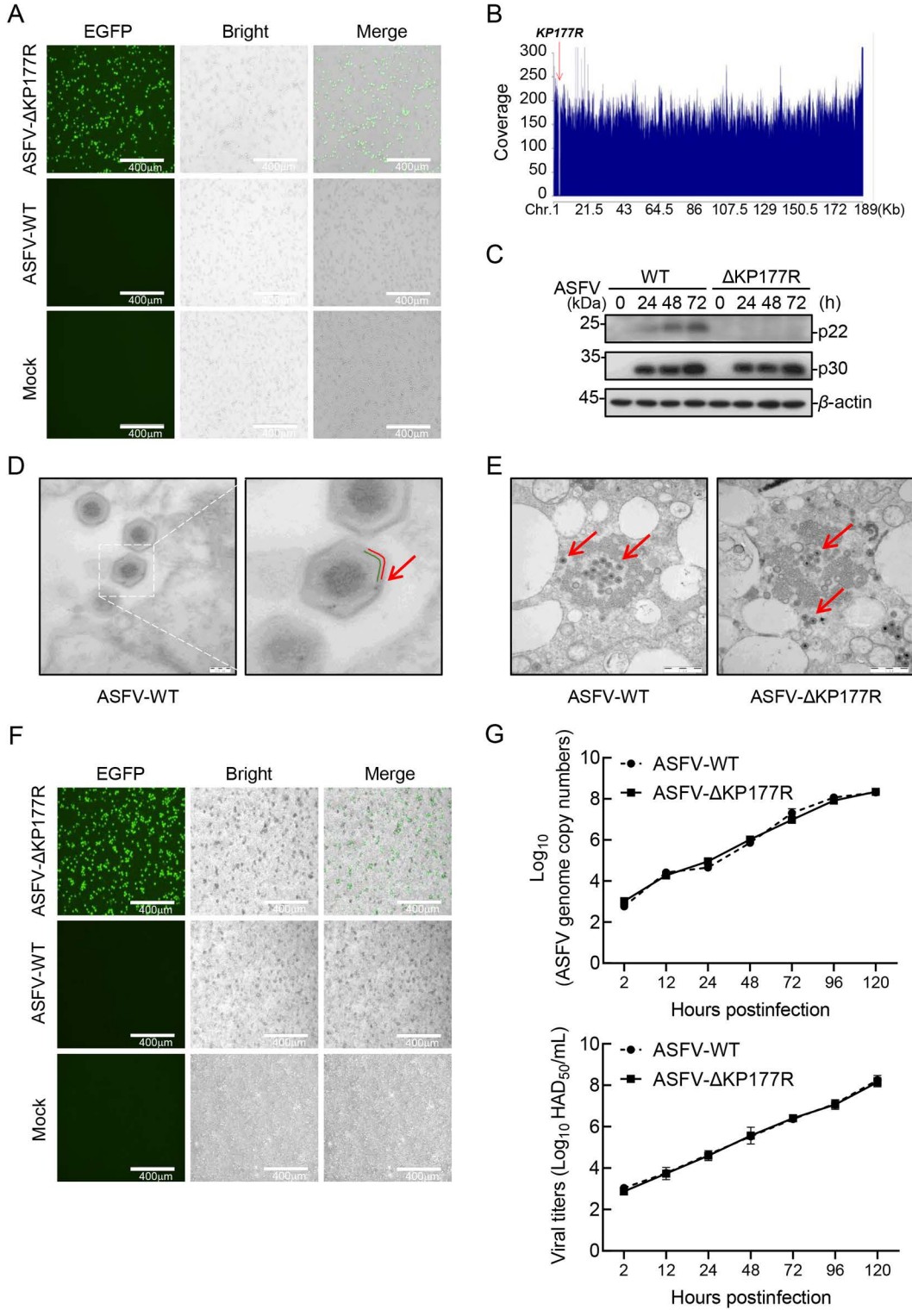

**Fig 1. The _KP177R_ gene is not essential for ASFV replication in porcine alveolar macrophages (PAMs).** (A) The recombinant virus ASFV-ΔKP177R was analyzed using direct fluorescence analysis. (B) Gene sequencing analysis of the viral genome. (C) PAMs were infected with ASFV-WT or ASFV-ΔKP177R (MOI = 0.1) for the indicated hours, and western blotting analysis was performed. (D) ASFV-WT-infected PAMs were fixed and

immunolabeled with a mouse antibody against p22 followed by an anti-mouse antibody conjugated to 10-nm-diameter gold particles. The subviral localization of p22 (red arrows) was observed by immunoelectron microscopy. The gold particles present on intracellular viruses were associated with the inner envelope (green) beneath the outer protein capsid (red). (E) The morphology of the mature virions (red arrows) was observed in the ASFV-ΔKP177R- or ASFV-WT-infected PAMs (MOI = 3) using transmission electron microscopy. (F) PAMs were infected with ASFV-ΔKP177R or ASFV-WT (MOI = 0.1). Hemadsorption property of the viruses was examined using microscopy. (G) PAMs were seeded into 24-well plates and infected with ASFV-ΔKP177R or ASFV-WT (MOI = 0.01). At the indicated time points, the genome copies and viral titers were determined using RT-qPCR and hemadsorption assays, respectively.

Next, the ability of ASFV-ΔKP177R to regulate the activation of the JAK-STAT signaling pathway was examined to verify the function of p22. Firstly, to exclude the potential influence of extracellular IFNs in the supernatant on JAK-STAT signaling, the intracellular mRNA levels of type I IFNs in PAMs infected with either ASFV-WT or ASFV-ΔKP177R were measured. The results showed that the deletion of the *KP177R* gene did not affect the mRNA transcription of *IFN-α* and *IFN-β* (S2B Fig). Subsequently, to corroborate the findings of the transcriptome analysis, the transcriptional levels of *ISGs* and several inflammatory cytokines were measured using RT-qPCR. The results revealed that the ASFV-ΔKP177R-infected PAMs had higher transcriptional levels of *STAT1*, interferon-induced protein with tetratricopetide repeats 1 (*IFIT1*), and radical S-adenosyl methionine domain-containing protein 2 (*RSAD2*) than did the ASFV-WT-infected PAMs (Fig 2C). Whereas, the deletion of p22 did not affect the transcriptional levels of chemokine ligand 2 (*CCL2*) and tumor necrosis factor alpha (*TNF-α*) (Fig 2D). Considering that the phosphorylation of STAT1 and STAT2 is the hallmark of the activation of the JAK-STAT signaling pathway [17,18], we further checked the effects of ASFV-ΔKP177R on their phosphorylation. The results showed that the ASFV-ΔKP177R-infected PAMs had higher levels of the phosphorylated STAT1 and STAT2 than did the ASFV-WT-infected PAMs (Fig 2E). These findings indicate that p22 plays an important role in immune evasion against the ASFV-triggered IFN responses.

## The ASFV p22 inhibits the IFN-β-triggered activation of the JAK-STAT signaling pathway

Considering the significance of p22 in the ASFV-triggered activation of the JAK-STAT signaling pathway, functional analysis of promoter activity, protein activation, and gene transcription were performed in the HEK293T and IBRS-2 cells *via* p22 overexpression (S2C Fig). The reporter assay results showed that p22 inhibited the IFN-β-triggered activation of the STAT1 promoter (Fig 3A), but not the TNF-α-triggered activation of the NF-κB promoter (Fig 3B), in a dose-dependent manner. The ectopically expressed p22 significantly inhibited the IFN-β-induced signaling, which is characterized by the phosphorylation of transcription factors, including STAT1 and STAT2 (Fig 3C). Consistent with these results, the transcription of *STAT1*, *IFIT1*, and *RSAD2* induced by IFN-β was remarkably inhibited by the ectopically expressed p22 (Fig 3D).

To verify the function specificity of p22 on JAK-STAT signaling pathway, the transcriptional levels of various proinflammatory cytokines were also measured in the p22-overexpressing HEK293T and IBRS-2 cells treated with TNF-α. The results showed that p22 had no effect on the TNF-α-triggered transcription of interleukin 8 *(IL-8)*, *CCL2*, and *TNF-α* (Fig 3E). Meanwhile, the role of p22 in IFN-γ- triggered JAK-STAT signaling pathway was also detected. The results showed that p22 slightly affected the IFN-γ-triggered transcription of the transcriptional levels of Guanylate-Binding Protein 1 (*GBP1*), CXC chemokineligand-10 (*CXCL10*), and the phosphorylation of STAT1 in the p22-overexpressing HEK293T cells (S2D and S2E Fig). These results demonstrate that p22 negatively regulates the IFN-β-triggered activation of the JAK-STAT signaling pathway.

## The ASFV p22 targets IFNAR1 and promotes its degradation

To elucidate the functional targets of p22, key molecules of the JAK-STAT signaling pathway (including IFNAR1, IFNAR2, JAK1, tyrosine kinase 2 [TYK2], STAT1, STAT2 and IRF9) were coexpressed with p22 in HEK293T cells. The coimmunoprecipitation (co-IP) assay results showed that p22 interacted with IFNAR1 specifically (Figs 4A and S3A). Consistent with

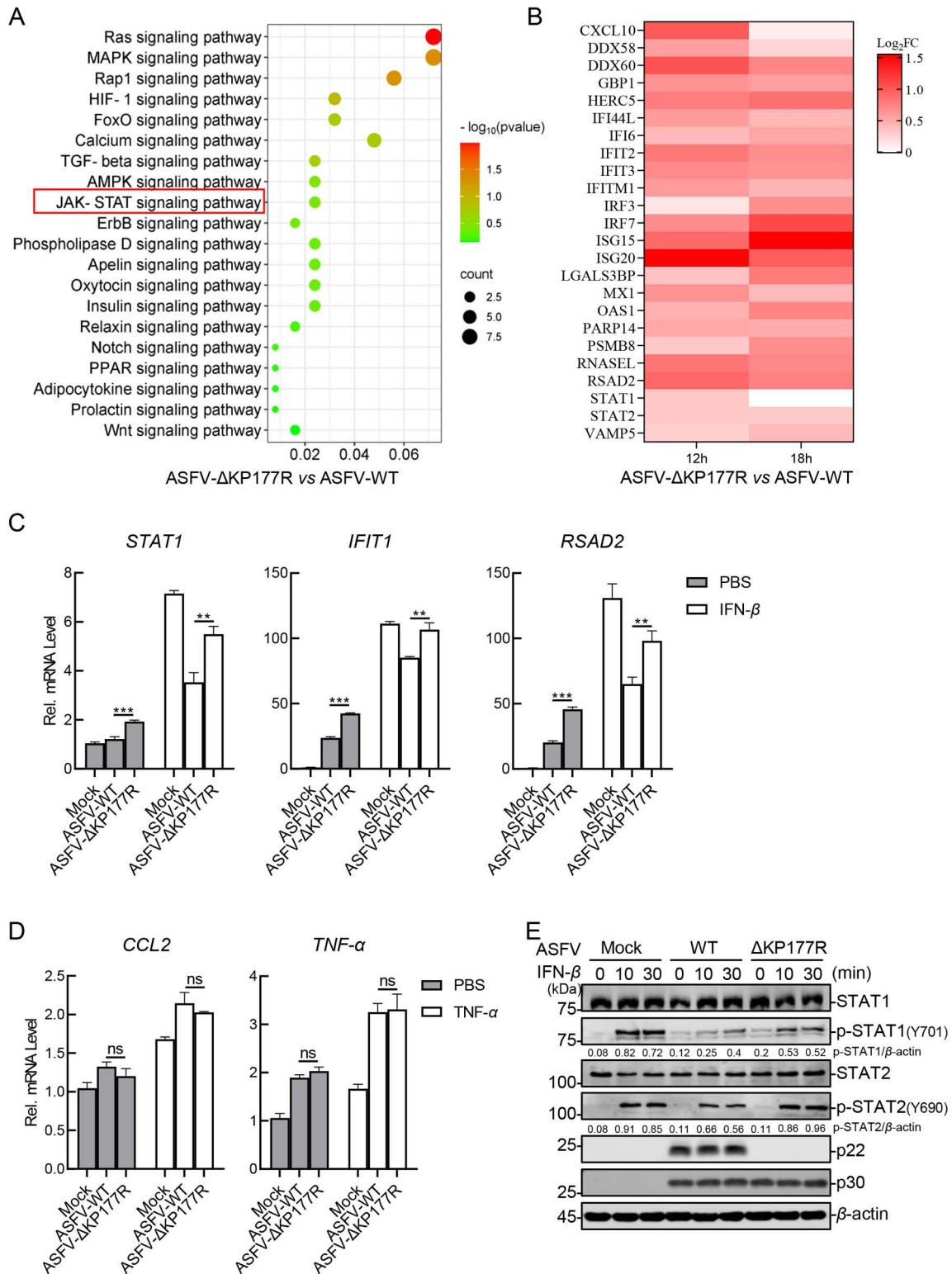

**Fig 2. The KP177R-deleted ASFV mutant significantly promotes the activation of the JAK-STAT signaling pathway in PAMs.** (A) The Kyoto Encyclopedia of Genes and Genomes (KEGG) enrichment analyses were performed in the ASFV-ΔKP177R- *vs* ASFV-WT-infected PAMs (MOI = 3). (B) Heat map of the *ISGs* in the ASFV-ΔKP177R- *vs* ASFV-WT-infected PAMs (MOI = 3). (C) PAMs were infected with ASFV-WT or ASFV-ΔKP177R

(MOI = 3). At 6 hpi, the cells were treated with 400 ng/mL IFN-β for 6 h. The total RNA was extracted using TRIzol reagent. Subsequently, the mRNA transcription of indicated genes was examined by RT-qPCR. (D) PAMs were infected with ASFV-WT or ASFV-ΔKP177R (MOI = 3). At 6 hpi, the cells were treated with 20 ng/mL TNF-α for 6 h, and mRNA transcription of indicated genes was determined as described above. (E) PAMs were infected with ASFV-WT or ASFV-ΔKP177R (MOI = 3). The PAMs were either treated with IFN-β at 12 hpi, and western blotting analysis was performed using the indicated antibodies.

the above-mentioned results, p22 was successfully detected in the ASFV-WT-infected PAMs and was able to bind to the endogenous IFNAR1, whereas the association of ASFV virions with IFNAR1 disappeared in the ASFV-ΔKP177R-infected PAMs (Fig 4B). Confocal microscopy showed that p22 was colocalized with IFNAR1 in IBRS-2 cells (Fig 4C), providing direct evidence for the interaction between IFNAR1 and p22.

Furthermore, we found that the presence of p22 resulted in a reduction in the protein levels of IFNAR1. To further confirm this interesting discovery, the plasmids expressing p22, IFNAR1, or IFNAR2 were cotransfected into HEK293T cells. Western blotting analysis showed that p22 overexpression indeed led to a significant decrease in the expression of IFNAR1, but not IFNAR2, in a dose-dependent manner (Figs 4D and S3B). On the other hand, the ectopically expressed p22 had no effect on the transcriptional level of *IFNAR1* (S3C Fig). Taken together, these data indicate p22 mediates the targeted degradation of overexpressed IFNAR1. Next, the influence of p22 on the endogenous IFNAR1 level was determined *via* p22 overexpression and viral infection. We found that the ectopically expressed p22 triggered the degradation of IFNAR1 in a dose-dependent manner (Fig 4E). Consistent with these results, the expression of IFNAR1 decreased upon infection with ASFV-WT, but not with ASFV-ΔKP177R (Fig 4F). In summary, these results show that p22 specifically targets IFNAR1 and facilitates its degradation.

## The ASFV p22 degrades IFNAR1 *via* autophagy

The ubiquitin-proteasome and lysosome pathways represent two prominent mechanisms of protein degradation in eukaryotic cells [36]. To explore the specific pathways involved in p22-mediated degradation of IFNAR1, HEK293T cells transfected with plasmids encoding IFNAR1 and p22 were treated with dimethyl sulfoxide (DMSO, control), carbobenzoxy-L-leucyl-L-leucyl-L-leucinal (MG132, a proteasome inhibitor), or chloroquine (CQ, a lysosome inhibitor). As depicted in Fig 5A, MG132 treatment could not rescue the expression of IFNAR1, whereas treatment with CQ effectively restored IFNAR1 level. Additionally, 3-methyladenine (3-MA, a phosphatidylinositol-3-hydroxykinase inhibitor) and SBI-0206965 (SBI, a UNC-51-like autophagy-activating kinase 1 [ULK1] inhibitor) inhibited the p22-mediated degradation of IFNAR1 in a dose-dependent manner in HEK293T cells (S4A and S4B Fig). Moreover, the ectopically expressed p22-mediated degradation of endogenous IFNAR1 was also restored upon treatment with CQ, 3-MA, or SBI (Fig 5B). These results suggest that inhibition of autophagy can compensate the degradation of IFNAR1 by p22.

Autophagy occurs as a result of a series of signaling cascades, in which the production of the microtubule-associated protein light chain 3 II (LC3-II) and the degradation of p62 are the hallmarks of the activation. To confirm the effect of p22 on the autophagy pathway, the expression levels of LC3-II and p62 were examined. The overexpression of p22 resulted in the upregulation of LC3-II and the degradation of IFNAR1 and p62 (Fig 5C), which suggests that p22 triggers the activation of autophagy. Pearson's correlation coefficients (R) were calculated to quantify the degree of colocalization between blue-labeled TAX1BP1 and green-labeled LC3. Consistently, p22 overexpression significantly enhanced the formation of EGFP-LC3 spots and facilitated the colocalization of IFNAR1 with EGFP-LC3 in IBRS-2 cells (Fig 5D). Furthermore, the PAMs infected with ASFV-WT or ASFV-ΔKP177R were subjected to western blotting analysis to assess the expression level of LC3-II. The results demonstrated that the expression level of LC3-II in the ASFV-ΔKP177R-infected PAMs was significantly lower than that in the ASFV-WT-infected PAMs (Fig 5E), suggesting that p22 induces autophagy.

Autophagy-related protein (ATG) 5 and ATG7 are key factors in autophagy, which participate in two ubiquitin-like conjugation systems and the elongation of autophagosomes during classical autophagy [37]. Subsequently, *ATG5* and *ATG7*

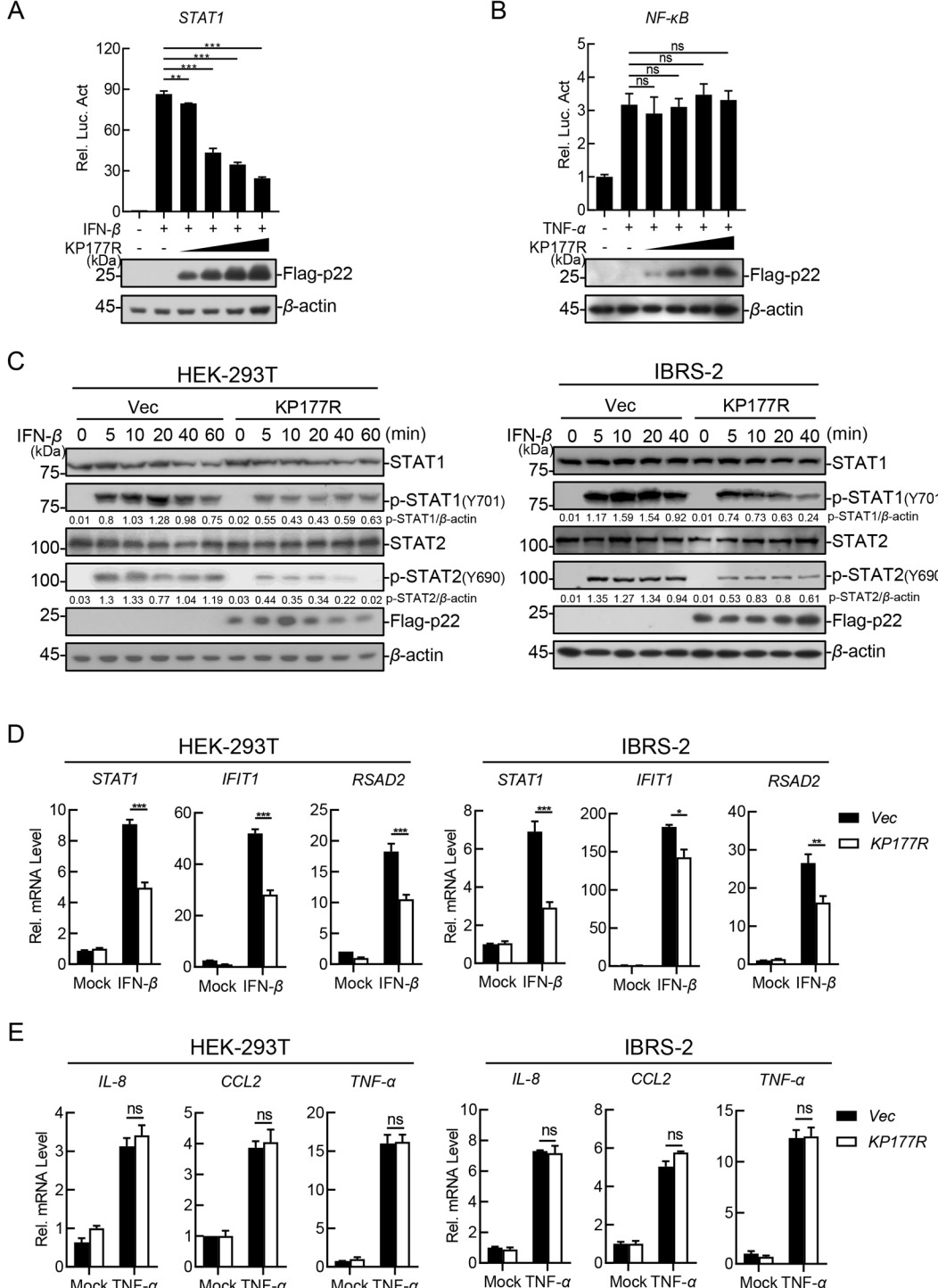

**Fig 3. The ASFV p22 inhibits the IFN-β-triggered activation of the JAK-STAT signaling pathway.** (A) HEK293T cells were cotransfected with different amounts of the Flag-p22-expressing plasmid and pSTAT1-Fluc. At 24 hours posttransfection (hpt), the cells were treated with IFN-β, followed by luciferase assay and western blotting analysis. (B) HEK293T cells were cotransfected with the different amounts of the Flag-p22-expressing plasmid, pNF-κB-Fluc, and pRL-TK. At 24 hpt, the cells were treated with 20 ng/mL TNF-α and subjected to luciferase assay and western blotting analysis. (C) HEK293T cells transfected with pFlag-KP177R or the IBRS-2 cells stably expressing p22 were treated with 200 ng/mL IFN-β for the indicated hours. Western blotting analysis was performed using the indicated antibodies. (D) HEK293T cells transfected with pFlag-KP177R or IBRS-2 cells stably

expressing p22 were treated with IFN-*β*. The cells were subjected to isolation of cellular total RNAs followed by quantification of transcription level of indicated genes by RT-qPCR. (E) HEK293T cells transfected with pFlag-KP177R or the IBRS-2 cells stably expressing p22 were treated with 20 ng/mL TNF-*α*, and the total RNA was extracted and analyzed using RT-qPCR as described above. The data were shown as the means ± SDs from one representative experiment performed in triplicates. * $P < 0.05$; ** $P < 0.01$; *** $P < 0.001$; and ns, not significant ($P > 0.05$) (unpaired Student's *t* test).

were knocked out or down to elucidate the indispensability of autophagy in the p22-mediated degradation of IFNAR1 [12]. Western blotting analysis showed that the p22-mediated degradation of IFNAR1 was inhibited once *ATG5* or *ATG7* was deleted (Figs 5F and S4C). These findings prove that p22 expression induces cellular autophagy and mediates the degradation of IFNAR1 *via* the autophagy pathway, and inhibition of cellular autophagy effectively counteracts the p22-mediated IFNAR1 degradation.

## TAX1BP1 is involved in the p22-induced degradation of IFNAR1

The autophagic pathways can be classified as either canonical or selective. ULK1, ATG13, and beclin-1 play crucial roles in the activation of the canonical autophagy pathway. Western blotting analysis showed that p22 did not interact with these three molecules (S5A Fig), which indicates that p22 functions *via* the selective autophagic pathway. Cargo receptors are responsible for facilitating the targeted delivery of cargo to autophagosomes. To identify the autophagy receptors accounting for IFNAR1 degradation, various autophagy receptor proteins (including NBR1, CALCOCO2, TAX1BP1, p62, OPTN, NIX, and TOLLIP) were coexpressed with p22 in HEK293T cells. The co-IP assay results showed that p22 interacted with TAX1BP1 but not the other receptors (Fig 6A), and that p22 and TAX1BP1 could pull each other down (Fig 6B). To determine whether p22 could interact with TAX1BP1 directly, we performed an *in vitro* GST pull-down assay using the prokaryotic-expressed GST-p22 fusion protein and eukaryotic HA-TAX1BP1 protein. The GST pull-down assay confirmed that p22 bound to TAX1BP1 *in vitro* (Fig 6C). Consistent with these results, confocal microscopy showed that p22 colocalized with TAX1BP1 in the cytoplasm (Fig 6D). Furthermore, p22 was detected successfully in the ASFV-WT-infected PAMs, and was found to be associated with the endogenous TAX1BP1, whereas this interaction was not observed in the ASFV-ΔKP177R-infected PAMs (Fig 6E).

Currently, the involvement of TAX1BP1 in the degradation of IFNAR1 remains largely unknown. Considering the observed interaction between p22 and TAX1BP1 in the abovementioned experiments, we wondered whether the overexpression of IFNAR1, TAX1BP1, and p22 in HEK293T cells could induce the degradation of IFNAR1 depending on TAX1BP1. Western blotting analysis showed that the overexpression of TAX1BP1 enhanced the p22-mediated degradation of the endogenous or ectopically expressed IFNAR1 in a dose-dependent manner (Figs 6F and S5B). Furthermore, TAX1BP1-knockout cell lines were generated, and selected the efficient KO-2 cell line was selected for subsequent experiments (S5C Fig). In the *TAX1BP1*-knockout cells, the p22-mediated degradation of the ectopically expressed IFNAR1 was completely blocked, and p22 expression was also significantly increased (S4C Fig). Similarly, *TAX1BP1* deletion also resulted in the upregulation of endogenous IFNAR1 and an increased in p22 expression, whereas p22 overexpression no longer facilitated the degradation of the endogenous IFNAR1 in the *TAX1BP1*-knockout cells (Fig 6G). As expected, the expression levels of IFNAR1 were significantly elevated in the *TAX1BP1*-knockout cells compared with those in the wild-type (WT) cells (Figs 6G and S5C). These findings suggest that TAX1BP1 plays an essential role in the degradation of IFNAR1, and TAX1BP1 is involved in the p22-induced degradation of IFNAR1.

## The ASFV p22 enhances the TAX1BP1-mediated degradation of IFNAR1

We observed remarkable upregulation of IFNAR1 in the *TAX1BP1*-knockout HEK293T cells compared with that in the WT cells (Figs 6G and S5C), suggesting that TAX1BP1 may directly mediate the degradation of IFNAR1. Western blotting analysis revealed that the overexpression of TAX1BP1 resulted in the degradation of IFNAR1. Furthermore, the degradation of IFNAR1 became even more pronounced with the presence of p22 (Fig 7A). Considering that TAX1BP1 is a

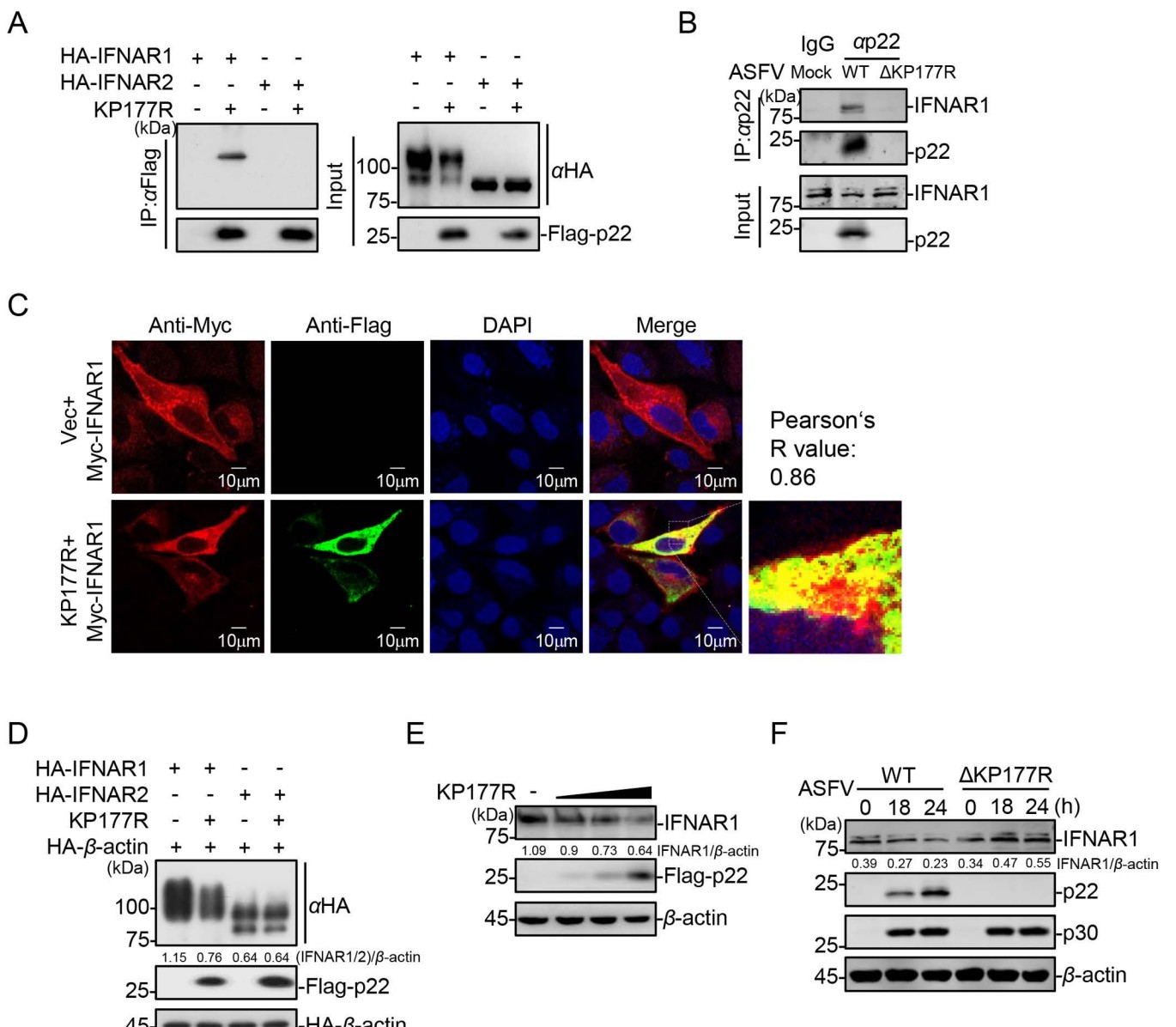

**Fig 4. The ASFV p22 targets IFNAR1 and promotes its degradation.** (A) HEK293T cells were cotransfected with the plasmids expressing HA-IFNAR1 or HA-IFNAR2, and Flag-p22 or the empty vector pRK (Vec) and then lysed and analyzed using co-IP with the anti-Flag MAb, followed by western blotting analysis using the indicated antibodies. (B) PAMs were infected with ASFV-WT or ASFV-ΔP (MOI = 3). The cells were lysed for co-IP using the anti-p22 MAb at 24 hpi, followed by western blotting analysis using the indicated antibodies. (C) IBRS-2 cells were cotransfected with the plasmids expressing Myc-IFNAR1 and Flag-p22 and then fixed with 4% paraformaldehyde. IFNAR1 and p22 were immunoblotted using anti-Myc and anti-Flag antibodies, respectively. The nuclei were stained with DAPI and subjected to confocal microscopy. The colocalization of p22 and IFNAR1 was analyzed using the Coloc2 tool in the ImageJ software (National Institutes of Health, Bethesda, MD, USA), and they are represented as Pearson's R value (R). Scale bar = 10 μm. (D) HEK293T cells were cotransfected with the HA-β-actin-, Flag-p22-, and HA-IFNAR1- or HA-IFNAR2-expressing plasmid, followed by western blotting analysis using the indicated antibodies at 24 hpt. (E) HEK293T cells were transfected with different amounts of the Flag-p22-expressing plasmid, followed by western blotting analysis using the indicated antibodies at 24 hpt. (F) PAMs were infected with ASFV-WT or ASFV-ΔKP177R (MOI = 3). At 18 and 24 hpi, the cells were lysed to perform western blotting analysis using the indicated antibodies.

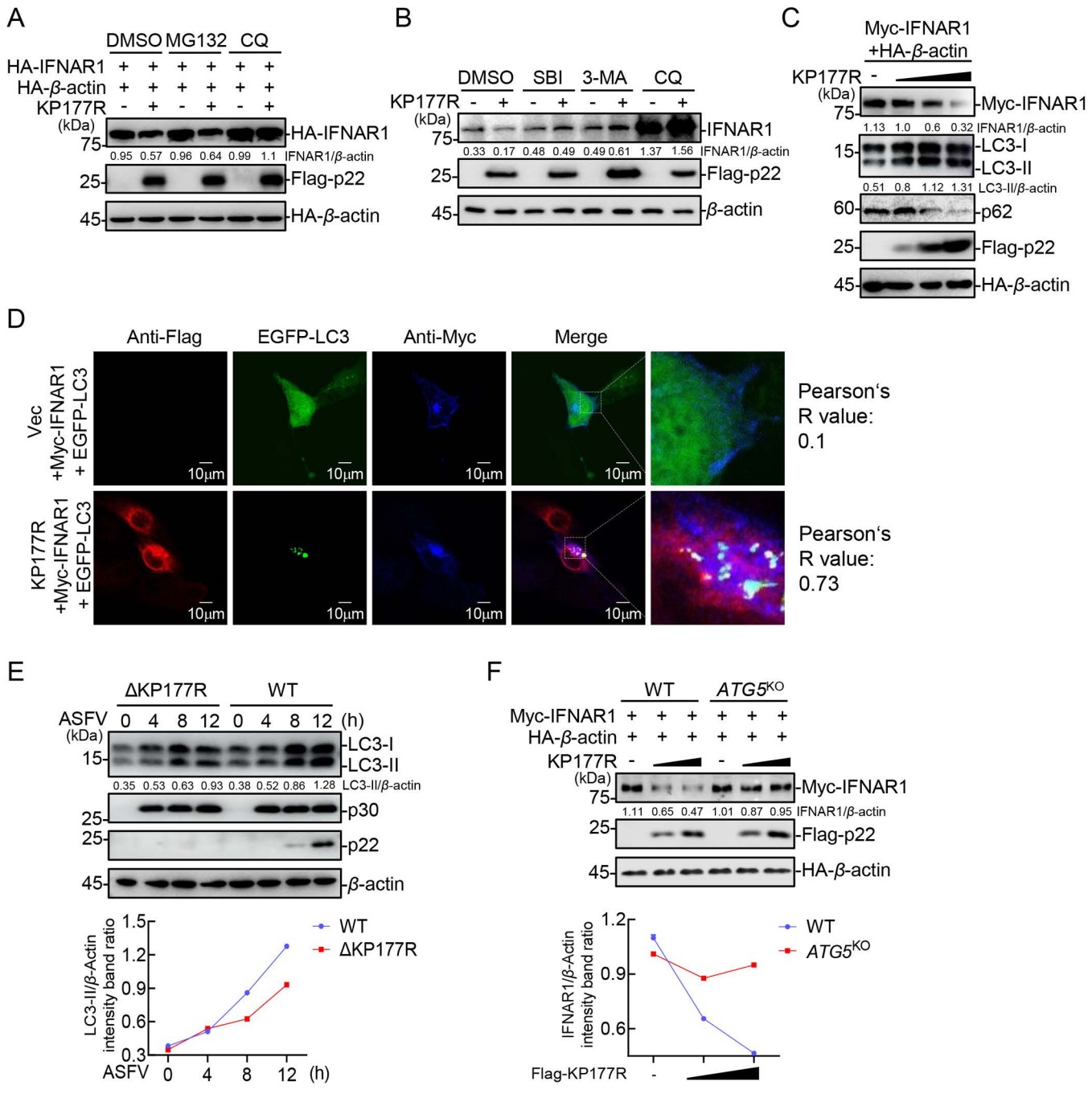

**Fig 5. The ASFV p22 degrades IFNAR1 *via* autophagy.** (A) HEK293T cells were cotransfected with the plasmids expressing HA-IFNAR1, HA-β-actin, and Flag-p22 or pRK (Vec). Then, the cells were treated with DMSO, MG132, or CQ at 24 hpt, followed by western blotting analysis using the indicated antibodies. (B) HEK293T cells were transfected with the plasmids pFlag-KP177R or pRK (Vec). Then, the cells were treated with DMSO, SBI, 3-MA, or CQ at 24 hpt, followed by western blotting analysis using the indicated antibodies. (C) HEK293T cells were transfected with the different amounts of the Flag-p22-expressing plasmid with pHA-IFNAR1 and pHA-β-actin, followed by western blotting analysis using the indicated antibodies at 24 hpt. (D) IBRS-2 cells were cotransfected with the plasmids expressing EGFP-LC3, Myc-IFNAR1, and Flag-p22, and then fixed with 4% paraformaldehyde. IFNAR1 and p22 were immunoblotted using anti-Myc and -Flag antibodies, respectively. The nuclei were stained with DAPI and analyzed using confocal microscopy. The colocalization of LC3 and IFNAR1 was analyzed using the Coloc2 tool in ImageJ and are shown as Pearson's R value (R). Scale bar = 10 μm. (E) PAMs were infected with ASFV-WT or ASFV-ΔKP177R for the indicated duration (MOI = 3), followed by western blotting analysis using the indicated antibodies. (F) The ATG5-knockout HeLa cells or the WT HeLa cells were transfected with the different amounts of the Flag-p22-expressing plasmid with pHA-IFNAR1 and pHA-β-actin, followed by western blotting analysis using the indicated antibodies at 24 hpt. The densitometric analysis of the protein expression levels was performed using the ImageJ software.

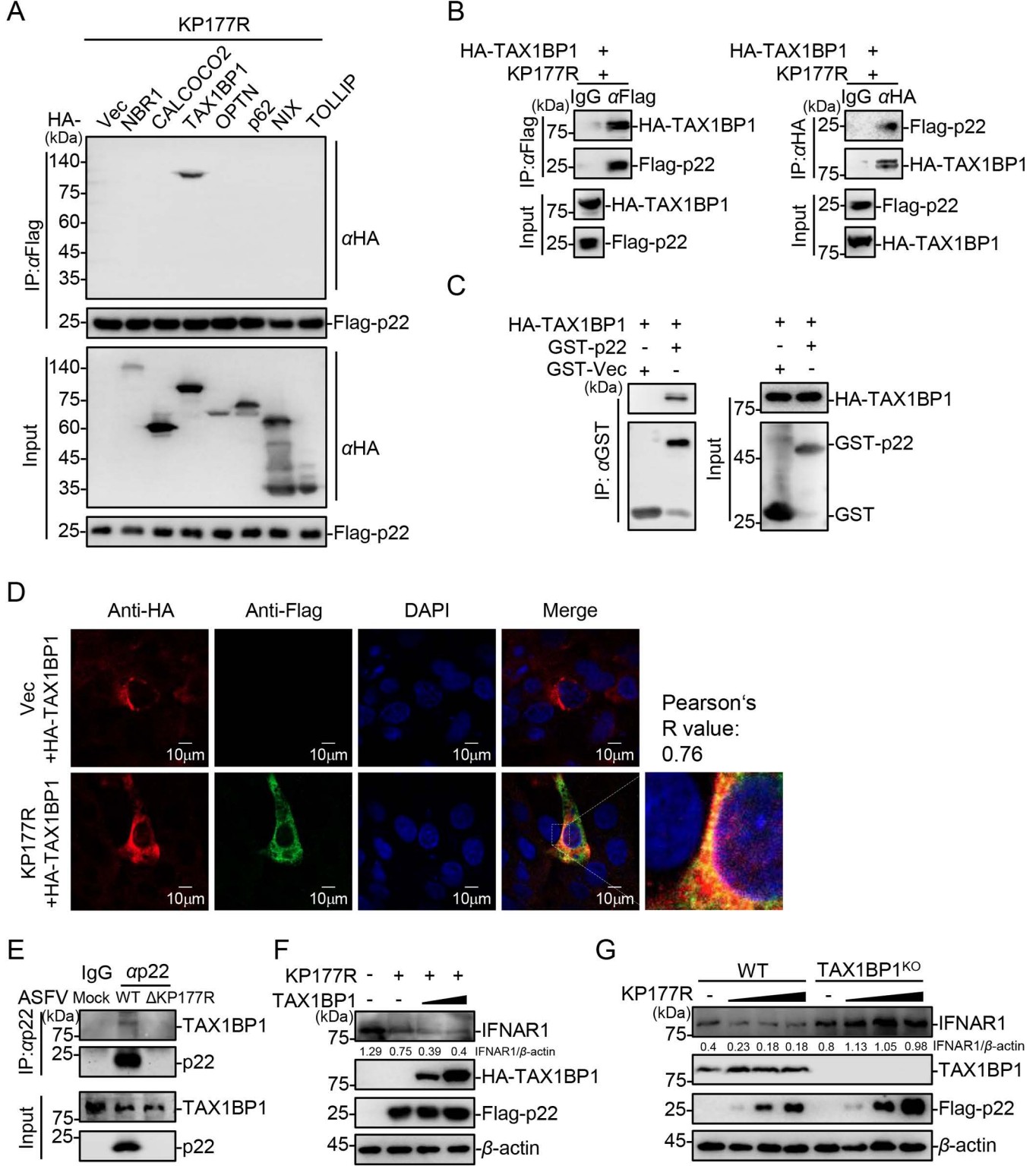

**Fig 6. TAX1BP1 is involved in the p22-induced degradation of IFNAR1.** (A) HEK293T cells were cotransfected with the plasmids expressing HA-NBR1, -CALCOCO2, -TAX1BP1, -p62, -OPTN, -NIX, or -TOLLIP, and Flag-p22, and then lysed for co-IP using the anti-Flag MAb, followed by western blotting analysis using the indicated antibodies. (B) HEK293T cells were cotransfected with the plasmids expressing Flag-p22 and HA-TAX1BP1,

and then lysed for Co-IP using anti-Flag or anti-HA MAb, followed by western blotting analysis using the indicated antibodies. (C) HEK293T cells were transfected with pHA-TAX1BP1, harvested and lysed at 24 hpt, and the supernatants were incubated with the prokaryotically expressed GST-p22 protein bound to the GST beads. The formed complexes were analyzed using western blotting and the indicated antibodies. (D) IBRS-2 cells were cotransfected with the plasmids expressing HA-TAX1BP1 and Flag-p22, and then fixed with 4% paraformaldehyde. TAX1BP1 and p22 were immunoblotted using anti-HA and anti-Flag antibodies, respectively. The nuclei were stained with DAPI and subjected to confocal microscopy. The colocalization of p22 and TAX1BP1 was analyzed using the Coloc2 tool in ImageJ, and it is shown as Pearson's R value (R). Scale bar = 10 μm. (E) PAMs were infected with ASFV-WT or ASFV-ΔKP177R (MOI = 3). At 24 hpi, the cells were lysed for co-IP using anti-p22 MAb, followed by western blotting analysis using the indicated antibodies. (F) HEK293T cells were transfected with different amounts of the HA-TAX1BP1-expressing plasmid with pFlag-KP177R, followed by western blotting analysis using the indicated antibodies at 24 hpt. (G) The TAX1BP1-knockout or WT HEK293T cells were transfected with different amounts of the Flag-p22-expressing plasmid or pRK (Vec), followed by western blotting analysis using the indicated antibodies at 24 hpt. The densitometric analysis of the protein expression levels was performed using the ImageJ software.

selective autophagy receptor, we used the autophagy inhibitors SBI, 3-MA, and CQ to explore whether TAX1BP1 directly mediates IFNAR1 degradation *via* cellular autophagy. Western blotting analysis revealed that TAX1BP1-mediated degradation of IFNAR1 were effectively inhibited under treatment with SBI, 3-MA, or CQ (Fig 7B).

Selective autophagy is generally divided into two types: ubiquitin-independent and ubiquitin-dependent [36]. As a selective autophagy receptor, TAX1BP1 contains a C-terminal ubiquitin-binding domain (UBD) with two zinc finger structures [38,39]. Structural analysis using online prediction software revealed that amino acids (aa) 645–706 constitutes the UBD of TAX1BP1 (http://smart.embl-heidelberg.de). Thus, a mutant lacking the UBD of TAX1BP1 (TAX1BP1-ΔUBD) was constructed to explore whether the function of TAX1BP1 depends on its ubiquitin-binding activity (Fig 8A). Our results demonstrated that TAX1BP1-ΔUBD retained the ability to promote the degradation of IFNAR1 in HEK293T cells (Fig 7C). Consistent with these findings, both TAX1BP1-ΔUBD and WT TAX1BP1 exhibited comparable promotion of the p22-mediated degradation of IFNAR1 (Fig 7D), which indicates that the ubiquitin-binding activity of TAX1BP1 is not essential for its degradation of IFNAR1. Considering that TAX1BP1 functions independently of its UBD domain, we hypothesized that the ubiquitination of IFNAR1 might remain unaffected by p22. To figure out the role of ubiquitination in p22 regulating the TAX1BP1-mediated degradation of IFNAR1, we cotransfected HEK293T cells with the plasmids expressing HA-Ub, Myc-IFNAR1, and Flag-p22. Western blotting analysis showed that IFNAR1 could be ubiquitinated, whereas p22 had no effect the ubiquitination of IFNAR1 (Fig 7E).

Subsequently, a binding assay based on co-IP was conducted to explore the underlying mechanism by which p22 promotes TAX1BP1-mediated degradation of IFNAR1. Our results showed that TAX1BP1 interacted with IFNAR1, and p22 enhanced the binding of TAX1BP1 to IFNAR1 in a dose-dependent manner (Fig 7F). These results suggest that TAX1BP1 regulates the degradation of IFNAR1 independently of its ubiquitin-binding activity, and p22 promotes the degradation of IFNAR1 by enhancing the binding of TAX1BP1 to IFNAR1.

### The transmembrane domain of the ASFV p22 is essential for inhibiting the activation of the JAK-STAT signaling pathway

To identify the crucial domain for the interaction of p22 with IFNAR1 and TAX1BP1, the structures of p22, IFNAR1, and TAX1BP1 were analyzed using an online prediction software (http://smart.embl-heidelberg.de). p22 contains an intracellular domain (ICD, aa 1–6), a transmembrane domain (TMD, aa 7–29), and an extracellular domain (ECD, aa 30–177). The transmembrane protein IFNAR1 comprises an ECD (aa 1–436), a TMD (aa 437–460), and an ICD (aa 461–560). TAX1BP1 has three domains: an N-terminal SKIP carboxyl homology domain (SKICH, aa 1–424), a central oligomerization domain containing a coiled-coil region (CC, aa 425–644), and a UBD containing two zinc finger motifs (aa 645–706). Based on the structural characteristics of p22, IFNAR1, and TAX1BP1, a series of mutants were constructed, listed in Fig 8A. Co-IP analysis showed that the TMD of p22 was responsible for its interaction with IFNAR1 and TAX1BP1 (S6A and S6B Fig), while IFNAR1 bound to p22 *via* its TMD (S6C Fig), and the CC region of TAX1BP1 was essential for its association with p22 (S6D Fig).

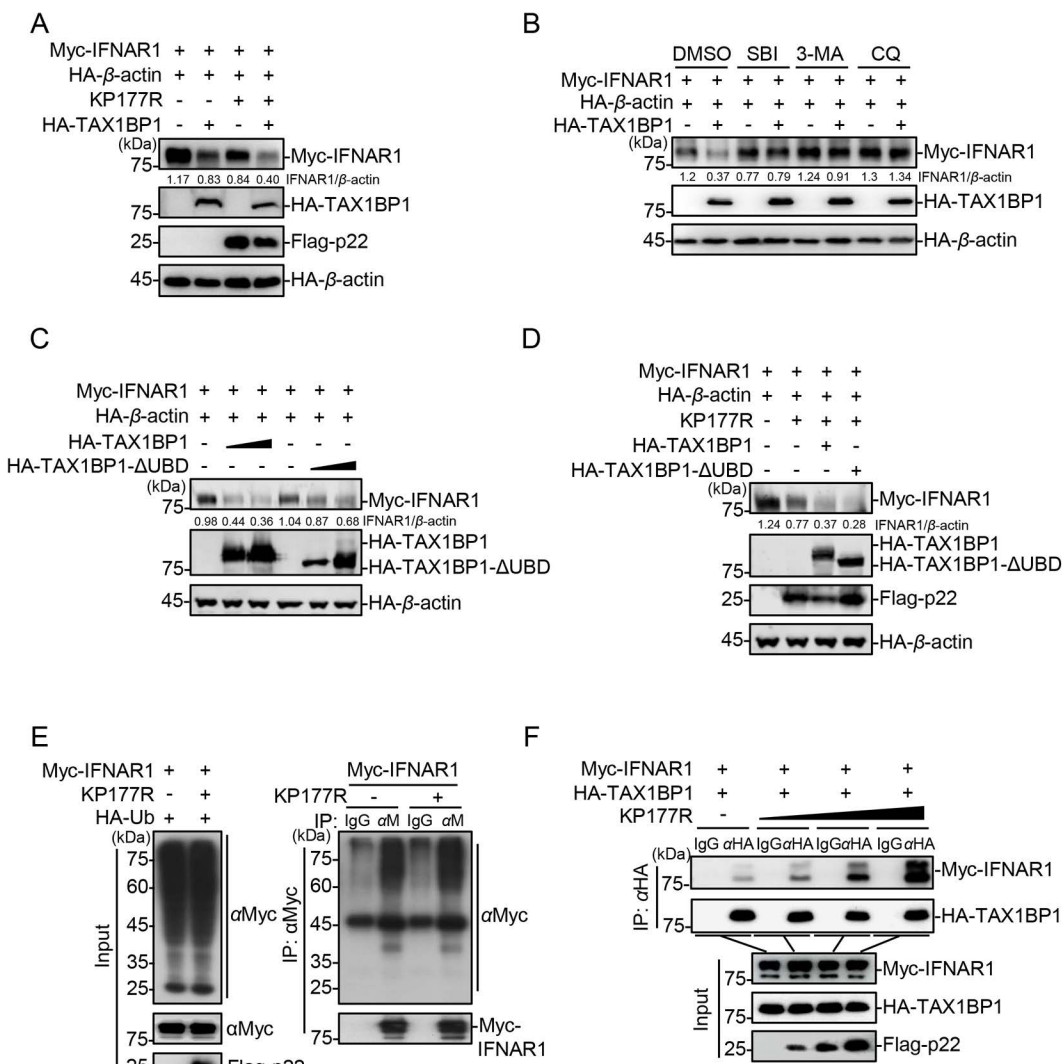

**Fig 7. p22 enhances the TAX1BP1-mediated degradation of IFNAR1.** (A) HEK293T cells were cotransfected with the plasmids expressing the Myc-IFNAR1, HA-β-actin, HA-TAX1BP1, and Flag-p22, followed by western blotting analysis using the indicated antibodies. (B) HEK293T cells were cotransfected with the plasmids expressing Myc-IFNAR1, HA-β-actin, and HA-TAX1BP1. Then, the cells were treated with DMSO, SBI, 3-MA, or CQ at 24 hpt, followed by western blotting analysis using the indicated antibodies. (C) HEK293T cells were transfected with different amounts of the TAX1BP1- or TAX1BP1-ΔUBD-expressing plasmids with pMyc-IFNAR1 and pHA-β-actin, followed by western blotting analysis using the indicated antibodies at 24 hpt. (D) HEK293T cells were cotransfected with the plasmids expressing Myc-IFNAR1, HA-β-actin, Flag-p22, and HA-TAX1BP1 or HA-TAX1BP1-ΔUBD, followed by western blotting analysis using the indicated antibodies. (E) HEK293T cells were cotransfected with the plasmids expressing Myc-IFNAR1, HA-Ub, and Flag-p22, and then lysed for co-IP using anti-Myc MAb, followed by western blotting analysis using the indicated antibodies. (F) HEK293T cells were transfected with different amounts of the p22-expressing plasmid with pMyc-IFNAR1 and pHA-TAX1BP1, and then lysed for co-IP using anti-HA MAb at 24 hpt, followed by western blotting analysis using the indicated antibodies.

The structure of a protein specifies its function. As our experimental results showed the importance of the TMD of p22 in interacting with IFNAR1 and TAX1BP1, we hypothesized that the TMD of p22 may also be the critical domain in the inhibition of the activation of the JAK-STAT signaling pathway by p22. First, the reporter assay showed that the mutant p22-ΔECD, with deletion of the ECD, was sufficient to remarkably inhibit the IFN-β-triggered activation of the STAT1 promoter, whereas mutants with deletion of the TMD exhibited the loss of its effect on the activation of the STAT1

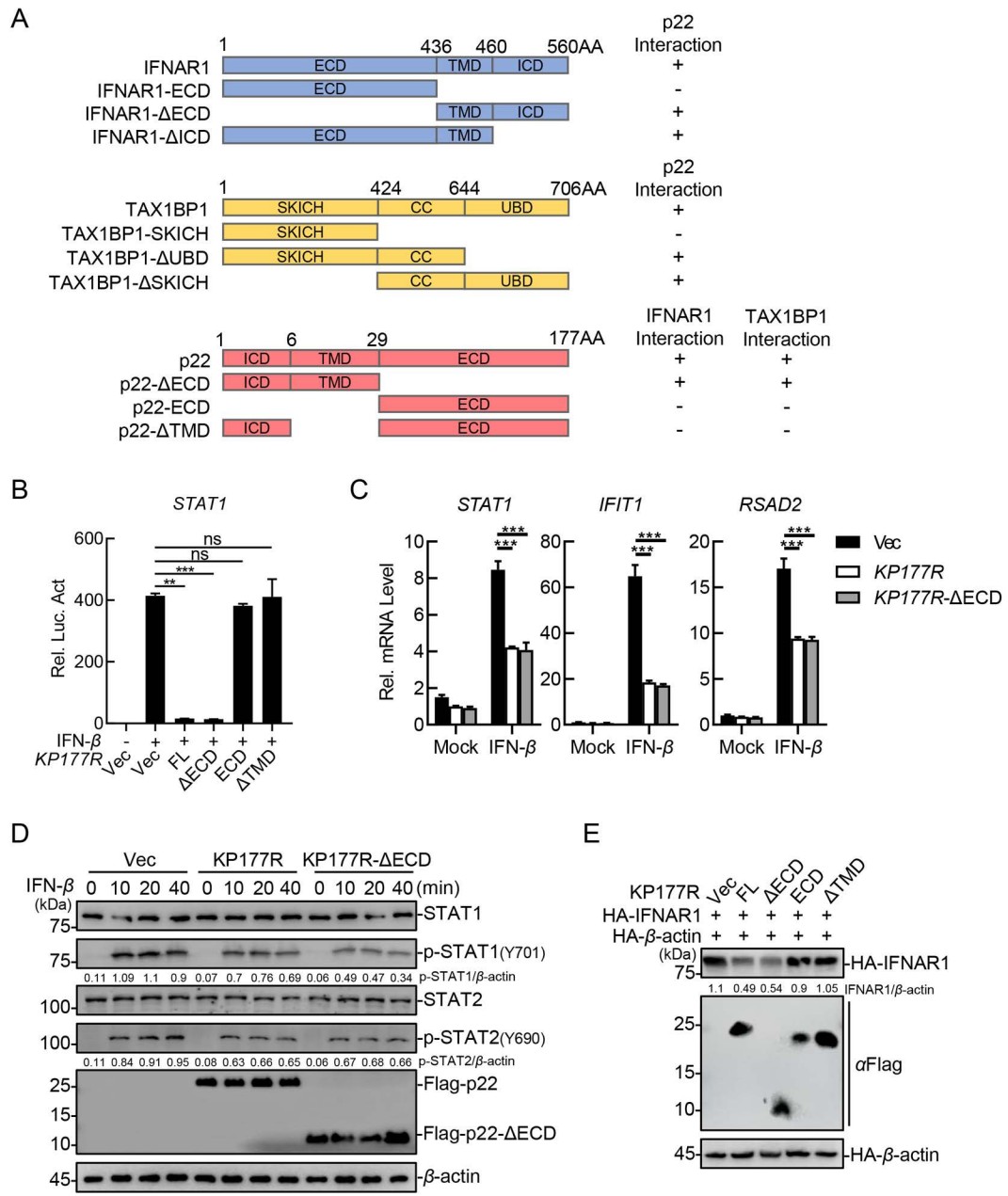

**Fig 8. The transmembrane domain of the ASFV p22 is essential for inhibiting the activation of the JAK-STAT signaling pathway.** (A) Schematic illustration of the domain structures of p22, IFNAR1, and TAX1BP1, and the construction strategy of their truncation mutants. (B) HEK293T cells were cotransfected with plasmids pSTAT1-Fluc and pFlag-KP177R or plasmids expressing the truncated versions at 24 hpt and luciferase assay was conducted. (C) IBRS-2 cells were transfected with pFlag-KP177R or pFlag-KP177R-ΔECD, and treated with 200 ng/mL IFN-β, followed by total RNA extraction and the subsequent RT-qPCR analysis. (D) IBRS-2 cells were transfected with pFlag-KP177R or pFlag-KP177R-ΔECD and treated with 200 ng/mL IFN-β for the indicated hours and subjected to western blotting analysis using the indicated antibodies. (E) HEK293T cells were cotransfected with the plasmids expressing Myc-IFNAR1, HA-β-actin, and Flag-p22 or the truncated versions of p22 and subjected to western blotting analysis using the indicated antibodies. The densitometric analysis of the protein expression levels was performed using the ImageJ software.

promoter (Fig 8B). The ectopic expression of p22-ΔECD or p22 remarkably inhibited the transcription of *ISGs*, such as *STAT1*, *IFIT1*, and *RSAD2,* in IBRS-2 cells (Fig 8C). Consistent with these results, western blotting analysis showed that p22-ΔECD, as well as p22, significantly inhibited the phosphorylation of STAT1 and STAT2 in IBRS-2 cells (Fig 8D). Furthermore, the extent of IFNAR1 degradation caused by different truncated p22 mutants was analyzed, and we found that p22-ΔECD, but not the other truncated p22 mutants, induced the degradation of IFNAR1 (Fig 8E). These findings suggest that the transmembrane domain of the ASFV p22 is essential for its interaction with IFNAR1 and TAX1BP1 and its inhibition of the JAK-STAT signaling pathway.

## Discussion

ASFV is a highly pathogenic agent, and its typical clinical symptoms include high fever, cyanosis, hemorrhagic lesions, anorexia, and ataxia, with a mortality rate of up to 100% for acute infection, which has led to significant economic losses in the global swine industry [40,41]. In ASFV, some structural proteins are involved in viral entry, such as p12, pE248R, and pE199L, and some are required for the assembly process, such as p17 and pE183L. Whereas, the functions of some structural proteins involved in ASFV replication remain unclear till now [42]. p22 is one such protein that is yet to be deciphered despite its significance. Proteomic analysis shows that the potential interacting proteins of p22 are involved in cellular metabolism, gene transcription, and autophagy [33]. Thus, the function of p22 in viral replication and further research on its underlying molecular mechanism are urgently needed to help prevent and control the spread of the disease.

In the struggle between the virus and its host, type I IFNs are sensed by the receptor complex IFNAR1/IFNAR2, followed by the activation of the JAK-STAT signaling pathway and the transcription and expression of downstream *ISGs*. These *ISGs* exert regulatory control over cellular antiviral responses, thereby impeding all stages of viral proliferation [17–20]. Porcine IFNs exhibit inhibitory effects on the replication of various viruses, including ASFV, classical swine fever virus, porcine reproductive and respiratory syndrome virus, and Japanese encephalitis virus, both *in vitro* and *in vivo* [35,43–45]. To survive in the host, viruses have evolved multiple strategies to evade the host immune response. For instance, SARS-CoV-2 targets IFNAR1 in the JAK-STAT pathway and destabilizes IFNAR1 through ubiquitination, resulting in cellular desensitization to type I IFNs [46]. The Zika virus non-structural protein 55 and the ASFV pS273R interacts with STAT2 and recruits the E3 ubiquitin ligase, resulting in polyubiquitination and subsequently the proteasome-dependent degradation of STAT2, thereby facilitating evasion of the IFN antiviral signaling [47,48]. In this study, we identified a novel mechanism by which ASFV evades the host IFN response. The transcriptome analysis revealed potential involvement of p22 in the ASFV-triggered JAK-STAT signaling pathway. Additionally, ASFV-ΔKP177R significantly promoted activation of the JAK-STAT signaling pathway in PAMs by assessing the phosphorylation levels of key molecules within this pathway and the transcription levels of *ISGs*. Subsequently, we verified the role of p22 in regulating the host IFN response using HEK293T and IBRS-2 cells. Meanwhile, we found that p22 did not affect the activation of TNF-*α*-triggered canonical NF-*κ*B signaling pathway. Interestingly, pI10L, a homologous protein of p22, was found to specifically regulate the TNF-*α*- and IL-1*β*-triggered activation of the canonical NF-*κ*B signaling pathway [49,50]. These findings suggest that p22 family proteins functions variously in viral replication and immune evasion. Mechanistically, p22 facilitates the interaction between TAX1BP1 and IFNAR1, enhancing the TAX1BP1-mediated degradation of IFNAR1 *via* the autophagy pathway. Consequently, this impedes the activation of the JAK-STAT signaling pathway and reduces the host antiviral response. As a giant and complex DNA virus, ASFV encodes many proteins to counteract host IFN response, such as pS273R, pMGF360-9L, pH240R, pMGF505-7R, and pF778R [47,51–53]. These proteins exhibit varying levels of abundance across different stages of viral replication and interact with distinct molecular targets within the JAK-STAT pathway during viral infection. It is speculated that they orchestrate the modulation of host inflammation *via* finely-tuned cooperative mechanisms.

Meanwhile, we observed a significant increase in p22 expression upon *TAX1BP1* knockout, and overexpression of TAX1BP1 resulted in a decrease in p22 (Figs 6G and 7A). This suggests that p22 is also degraded *via* the autophagy

pathway after interacting with TAX1BP1 and IFNAER1. These findings contribute to our understanding of the function of p22 in ASFV immune evasion strategies. However, whether the pathogenicity and virulence of ASFV-ΔKP177R differs from ASFV-WT *in vivo* still needs further studies.

The intracellular stability of IFNAR regulates the antiviral innate immune response against pathogen invasion [21]. The extracellular ligands type I IFN or LDL receptor related protein associated protein 1 bind to IFNAR1 on the cell surface, promoting the phosphorylation of IFNAR1 and subsequently recruiting the E3 ubiquitin ligase β-TrCP2 for IFNAR1 ubiquitination and proteolysis [21,23]. Intracellularly, p38 protein kinase and casein kinase 1α facilitate the phosphorylation of IFNAR1 through their kinase activity, which subsequently triggers the ubiquitination of IFNAR1 by E3 ubiquitin ligase β-TRCP, leading to its degradation *via* the lysosomal pathway [54–56]. However, we found that the expression levels of IFNAR1 were notably elevated in the TAX1BP1-knockout cells compared with those in the WT cells. Further experiments showed that the autophagy receptor TAX1BP1 interacts with IFNAR1, leading to its degradation *via* the autophagic pathway. Ubiquitination is a common posttranslational modification that serves as a crucial degradation signal and provides recognition sites for numerous autophagy receptors. TAX1BP1 facilitates the delivery of cargoes to autophagosomes by binding to the ubiquitin chain of substrates *via* its UBD domain [36]. However, we discovered that TAX1BP1 can degrade the substrates independently of ubiquitin. The overexpression mutant TAX1BP1-ΔUBD regulates the degradation of IFNAR1 and promotes the p22-mediated degradation of IFNAR1. However, autophagic degradation independently of ubiquitination is not limited to TAX1BP1. For instance, the autophagy receptor NIX interacts with LC3 *via* its LC3-interacting region (LIR) to induce mitophagy [57,58]. These findings enrich our understanding of the mechanisms underlying the TAX1BP1-mediated selective autophagy. The present study elucidated a novel degradation mechanism of IFNAR1 independently of ubiquitination. These findings will facilitate the development of new strategies for the diagnosis and management of infectious diseases and autoimmune diseases.

In conclusion, our findings demonstrated a critical role of p22 in regulating the IFN-β-triggered JAK-STAT signaling by targeting IFNAR1 (Fig 9). We discovered that the selective autophagy receptor TAX1BP1 interacted with IFNAR1 and facilitated its degradation through the autophagy pathway. The transmembrane region of the ASFV p22 enhances the binding of TAX1BP1 to IFNAR1, thereby facilitating the degradation of IFNAR1. This further led to a decrease in the phosphorylation of key molecules in the JAK-STAT pathway and a decrease in the transcription of antiviral genes, ultimately inhibiting the host antiviral response. These findings enhance our understanding of the mechanisms underlying IFNAR1 degradation and offer a promising avenue for the development of drugs targeting IFNAR-related diseases. This study elucidated the function of p22 in regulating the IFN response after ASFV infection and provides a novel theoretical basis for understanding the biological characteristics of ASFV. This study will contribute to the development of vaccines and antiviral agents against ASFV.

## Materials and methods

### Ethics statement

PAMs were isolated from lung lavage fluid of 4-week-old healthy specific pathogen-free (SPF) pigs, and the animal experiment was conducted in compliance with the Animal Welfare Act and Guide for the Care and Use of Laboratory Animals, approved by the Laboratory Animal Welfare Committee of Harbin Veterinary Research Institute (HVRI) of the Chinese Academy of Agricultural Sciences (approval number 220602–01-GR).

### Biosafety statement and facilities

All the experiments with live ASFVs were conducted within the animal biosafety level 3 (ABSL-3) facilities in the Harbin Veterinary Research Institute (HVRI) of the Chinese Academy of Agricultural Sciences (CAAS).

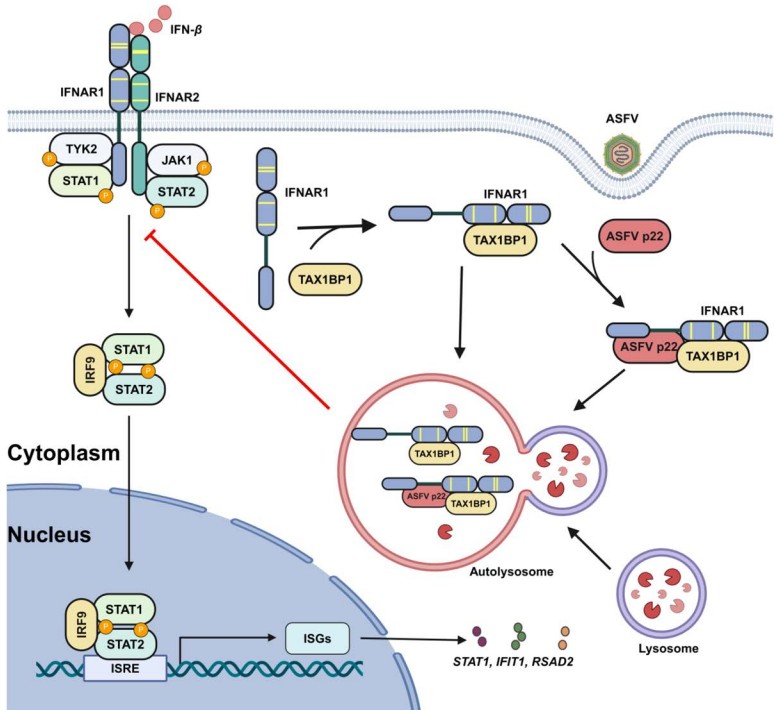

**Fig 9. Schematic diagram of the mechanism by which the ASFV p22 suppresses cellular IFN responses.** The selective autophagy receptor TAX1BP1 interacts with and degrades IFNAR1 *via* autophagy. The transmembrane region of the ASFV p22 enhances the binding of TAX1BP1 to IFNAR1, thereby promoting the degradation of IFNAR1. This leads to a decrease in the transcriptional levels of antiviral genes, ultimately inhibiting the host IFN responses. Created with BioRender.com.

## Cells and virus

PAMs were cultured in RPMI-1640 medium containing 10% fetal bovine serum (Gibco 10099–158, Grand Island, USA), 100 U/mL penicillin, 50 mg/mL streptomycin (Sigma-Aldrich, P0781, Germany). HEK293T cells were kindly provided by Prof. Hong-Bing Shu at Wuhan University. IBRS-2, a swine kidney cell line, were obtained from the American Type Culture Collection. HEK293T and IBRS-2 cells were cultured in Dulbecco's modified Eagle's medium supplemented with 10% fetal bovine serum. All cells were tested using specific primers to ensure the absence of conventional pathogen contamination. The ASFV HLJ/2018 strain (ASFV-WT) (GenBank accession no. MK333180.1) was isolated and propagated as described previously [59].

## Antibodies and reagents

The rabbit anti-STAT1 monoclonal antibodies (MAb) (14994), rabbit anti-P-STAT1 (Y701) MAb (9167), rabbit anti-STAT2 MAb (72604), rabbit anti-p-STAT2 (Y690) MAb (88410), and rabbit anti-ATG7 MAb (8558T) were purchased from Cell Signaling Technology (Danvers, USA). The rabbit anti-p62 MAb (ab109012) and rabbit anti-KDEL MAb (ab176333, endoplasmic reticulum) were sourced from Abcam. Rabbit anti-IFNAR1 polyclonal antibodies (PAb) (a1715), rabbit anti-β-actin MAb (AC026), rabbit anti-LC3B MAb (A19665), rabbit anti-LAMP1 pAb (A16894, lysosomes), rabbit anti-TOM20 pAb (A6774, mitochondria), rabbit anti-PSMA1 pAb (A3460, proteasomes), and rabbit anti-GM130 pAb (A11408, Golgi apparatus) were obtained from ABclonal (Wuhan, China). DyLight 488 goat anti-mouse IgG (A23210) and DyLight 594 goat anti-rabbit IgG (A23420) were procured from Abbkine (Wuhan, China). DyLight 405 goat anti-rabbit IgG (A0605) was acquired from Beyotime (Shanghai, China). Rabbit anti-HA tag PAb (51064–2-AP), mouse anti-HA tag MAb (66006–2-Ig),

mouse anti-Flag tag MAb (66008–4-Ig), rabbit anti-Myc tag PAb (16286–1-AP), mouse anti-GST tag MAb (66001–2-Ig), rabbit anti-TAX1BP1 PAb (14424–1-AP), HRP-conjugated goat anti-rabbit IgG(H+L) (SA00001–2), and HRP-conjugated goat anti-mouse IgG(H+L) (SA00001–1) were purchased from Proteintech (Chicago, USA). Mouse anti-ASFV-p22 MAb and mouse anti-ASFV-p30 MAb were produced and stored in our laboratory [60].

The mouse IgG (A7028) and rabbit IgG (A7016) were sourced from Beyotime. The anti-Flag M2 magnetic beads (M8823) were obtained from Sigma-Aldrich. ChromoTek GST-Trap Agarose was purchased from Proteintech. The protein A/G magnetic beads (PB101–02) were acquired from Vazyme (Nanjing, China). The recombinant human TNF-$\alpha$ (300-01A) and IFN-$\beta$ (300–02BC) were procured from PeproTech (Cranbury, NJ, USA). SBI-0206965 (HY-16966) was obtained from MedChemExpress (NJ, USA). MG132 (T2154) and 3-methyladenine (T1879) were purchased from TargetMol (Boston, USA). Chloroquine (S6999) was sourced from Selleck Chemicals (Houston, USA). DMSO (D8371) and 4',6-diamidino-2'-phenylindole (DAPI, C0065) was purchased from Solarbio (Beijing, China).

## Plasmids and transfection

The recombinant plasmids encoding HA-, Flag-, or Myc-tagged p22, IFNAR1, IFNAR2, JAK1, TYK2, STAT1, STAT2, IRF9, $\beta$-actin, NBR1, CALCOCO2, TAX1BP1, OPTN, P62, NIX, TOLLIP, ULK1, ATG13, and beclin-1, along with their mutants, were generated using standard molecular biology techniques. The plasmids pNF-$\kappa$B-Fluc, pSTAT1-Fluc, pRL-TK, psPAX2, and pMD2.0G, and the pLOV vector were also provided by Prof. Hong-Bing Shu. Among them, pSTAT1-Fluc and pRL-TK were originally purchased from QIAGEN, and pNF-$\kappa$B-Fluc was originally provided by Professor Gary Johnson at North Carolina State University, USA.

HEK293T cells were transfected using standard polyethylenimine (Polysciences, 24765–1) method. IBRS-2 cells were transfected using Lipofectamine 2000 (Invitrogen, 11668019). Chemically synthesized 21-nucleotide siRNA duplexes were acquired from GenePharma and transfected using Lipofectamine RNAiMAX (Invitrogen, 13778030). Where necessary, empty control plasmid was added to ensure that each transfection was performed using the same amount of the total DNA.

## Generation and identification of the KP177R-deleted ASFV mutant

Recombinant ASFV-ΔKP177R was generated *via* the homologous recombination method using the parental ASFV genome and a recombination transfer vector [34]. The recombinant transfer vector pOK12-p72-EGFP-ΔKP177R was constructed, containing genomic sequences flanking the targeted gene with approximately 1.2 kb upstream and downstream homologous arms, as well as an EGFP reporter controlled by the ASFV *B646L* gene promoter (The primers are listed in S2 Table). The left and right arms flanking the target gene were located at positions 2051–3250 and 3785–4984, respectively, within the ASFV-WT genome. The nucleotides within the genomic region spanning positions 3251–3784 were substituted with an expression cassette harboring the *EGFP* gene, resulting in the generation of the recombinant transfer vector pOK12-p72-EGFP-ΔKP177R. The ASFV-ΔKP177R mutant was generated through homologous recombination between the ASFV-WT genome and the recombination transfer vector using infection and transfection procedures in PAMs. Subsequently, viral purification was achieved by successive limiting dilutions of PAMs. The purified ASFV-ΔKP177R was then amplified in PAMs to obtain a viral stock. To validate the expected deletion of the target gene in each recombinant genome, the viral DNA and total proteins extracted from the ASFV-ΔKP177R-infected PAMs were subjected to sequencing and western blotting analysis.

## Hemadsorption (HAD) assay

The HAD assay was used to evaluate the median tissue culture infectivity of ASFV [61]. Briefly, PAMs ($5 \times 10^4$) were seeded into 96-well plates and infected with ASFV-ΔKP177R or ASFV-WT at an MOI of 0.1. At 48 hpi, fresh porcine red blood cells ($5 \times 10^5$) were added to each well. The red blood cells adsorbed onto the surface of ASFV-infected PAMs, forming rosette-like structures, and the "rosettes" of red blood cells were observed using an optical microscope.

## TEM assay

This experiment was performed as described previously [34]. Briefly, PAMs were seeded into 6-well plates and infected with ASFV-WT or ASFV-ΔKP177R at an MOI of 3, and the cells were harvested at 24 hpi and fixed with 2% glutaraldehyde in PBS for 1 h. The samples were dehydrated, embedded, and stained according to standard procedures. The samples were analyzed using an H-7650 (Hitachi, Tokyo, Japan) operated at 80 kV.

## Determination of ASFV-ΔKP177R replication kinetics

To evaluate the replication kinetics of ASFV-ΔKP177R, the growth curve assays were performed for ASFV-ΔKP177R and ASFV-WT in PAMs. Briefly, PAMs were seeded into 24-well plates and infected with ASFV-ΔKP177R or ASFV-WT at an MOI of 0.01 and rinsed twice with phosphate buffer saline (PBS) and replaced with fresh medium at 2 hpi. Subsequently, the supernatants of PAMs were harvested at 2, 12, 24, 48, 72, 96, and 120 hpi, and used to determine viral titers as $HAD_{50}$/mL in PAMs.

## Construction of stable expression cell lines and knockout cell lines

To create stable cells overexpressing p22 and the TAX1BP1-knocked cells, we employed lentivirus-mediated gene editing technology [50]. Briefly, HEK293T cells were cotransfected with the packaging plasmids psPAX2 and pMD2.0G, and either the donor plasmid pLOV-KP177R or the pLOV vector. At 48 hours post transfection (hpt), the recombinant viruses-containing culture supernatants were collected and used to transduce IBRS-2 cells in the presence of 8 μg/mL polybrene (Sigma-Aldrich; TR-1003-G). The stable cell lines were selected against 3 μg/mL puromycin (Solarbio, P8230) treatment for three passages of the transduced cells. Finally, the expression of p22 in these cells was verified by western blotting.

To create the TAX1BP1-knockout cells, HEK293T cells were cotransfected with the packaging plasmids psPAX2 and pMD2.0G, the donor plasmid LentiCRISPR-V2-TAX1BP1 or the LentiCRISPR-V2 vector. At 48 hpt, the recombinant viruses-containing culture supernatants were harvested and transduced into HEK293T cells in the presence of 8 μg/mL polybrene. The stable cell lines were selected against 0.5 μg/mL puromycin treatment for three passages of the transduced cells. Finally, the knockout of *TAX1BP1* in these cells was confirmed by western blotting. *ATG5*-knockout HeLa cells were previously reported [29].

## Dual-luciferase reporter assay

For the dual-luciferase reporter assay, HEK293T cells grown in 48-well plates were cotransfected with the reporter plasmids pNF-κB-Fluc (0.1 μg/well) and pRL-TK (0.01 μg/well), or pSTAT1-Fluc (0.01 μg/well), along with pFlag-KP177R or pRK (10, 20, 40, and 80 ng/well). At 20 hpt, the cells were treated with TNF-α or IFN-β for 10 h and then lysed using a passive lysis buffer. Luciferase activity was measured using a dual-specific luciferase assay kit (Promega, E1910). *Firefly* luciferase activity was normalized to *Renilla* luciferase activity. The data represent the means from three independent experiments.

## Reverse transcription-quantitative PCR (RT-qPCR)

The TRIzol reagent (TaKaRa, 9109) was employed for cellular total RNA extraction, which was then reverse-transcribed into cDNA using the HiScript III 1st strand cDNA synthesis kit (Vazyme, R312-01) following the manufacturer's protocols. RT-qPCR was performed using HiScript II Q RT SuperMix (Vazyme, R223-01) according to the manufacturer's protocols. The data are shown as relative mRNA normalized to GAPDH. The primers used for RT-qPCR are listed in S1 Table.

## Western blotting

Cell samples were collected by lysing with 2 × sodium dodecyl sulfate-polyacrylamide gel electrophoresis (SDS-PAGE) loading buffer, followed by denaturation at 95 °C for 20 min. Samples were separated by SDS-PAGE, transferred to

polyvinylidene fluoride membranes (PVDF) (Millipore, ISEQ00010) and subsequently blocked with 5% non-fat milk for 1 h. The detection of these proteins was accomplished using specific primary antibodies and HRP-labeled secondary antibodies. Subsequently, the protein was measured using enhanced chemiluminescence (Epizyme, SQ202L-2).

### Co-IP assay

For the co-IP assay, the cells were lysed with the M2 lysis buffer (20 mM Tris-HCl [pH 7.5], 0.5% NP-40, 10 mM NaCl, 3 mM EDTA, and 3 mM EGTA) containing protease inhibitors and sonicated for 2 min. Lysates were centrifuged at 13,000 $g$ for 10 min at 4 °C. The supernatants were subjected to immunoprecipitated with specific antibodies or anti-Flag M2 magnetic beads for 4 h. After three washes with the high-salinity M2 lysis buffer (0.5 M NaCl), the bound proteins were separated using SDS-PAGE, followed by western blotting analysis with the indicated antibodies, as described earlier.

### GST pull-down assay

For the GST pull-down assay, the GST and GST-p22 proteins were expressed in the prokaryotic system and subsequently purified by incubation with GST-Trap agarose beads for 4 h at 4 °C. GST and GST-p22 proteins were incubated with the lysates of the HEK293T cells containing the ectopically expressed HA-TAX1BP1 for 4 h at 4 °C. After three washes with the high-salinity M2 lysis buffer (0.5 M NaCl). Subsequently, western blotting analysis was performed using the indicated antibodies, as described earlier.

### Confocal microscopy

The IBRS-2 cells were seeded into coverslips in 24-well plates and subsequently transfected with the indicated plasmids. At 24 hpt, the IBRS-2 cells were fixed with 4% paraformaldehyde for 20 min and permeabilized using 0.1% triton X-100 for 15 min. Cells were blocked with 5% bovine serum albumin (BSA) for 30 min, and then incubated with the indicated antibodies and corresponding dye-conjugated secondary antibodies. Finally, the cells were stained with DAPI for 10 min. The images of the cells were obtained using a Zeiss confocal microscope under a 63 × oil-immersion lenses.

### Statistical analysis

The GraphPad Prism software version 8.0 (La Jolla, CA, USA) was used for statistical analysis. The quantitative data in histograms were shown as means ± standard deviations (SDs). The unpaired Student's $t$ test was used for data analysis. Statistical significance was set at $P < 0.05$. Asterisks in the figures indicate statistical significance: *, $P < 0.05$; **, $P < 0.01$; ***, $P < 0.001$.

### Supporting information

**S1 Fig. Subcellular localization analysis of ASFV p22.** PAMs were infected with ASFV (MOI = 1). At 24 hpi, the cells were fixed with 4% paraformaldehyde. The subcellular localization of p22 was determined by immunofluorescence using the indicated antibodies, and the nuclei were stained with DAPI and subjected to confocal microscopy. The colocalization of p22 and organelle markers was analyzed using the Coloc2 tool in ImageJ, and the colocalization results represented as Pearson's R value. Scale bar = 5 μm.
(TIF)

**S2 Fig. The ASFV p22 specificity negatively regulates the IFN-$\beta$-triggered activation of the JAK-STAT signaling pathway.** (A) PAMs were infected with ASFV-WT or ASFV-ΔKP177R (MOI = 3). At 12 hpi, the total RNA was extracted using TRIzol reagent. Subsequently, the mRNA transcription of indicated genes was examined by RT-qPCR. (B) PAMs were infected with ASFV-WT or ASFV-ΔKP177R (MOI = 1). At 24 hpi, the cells were subjected to isolation of cellular total RNAs followed by quantification of transcription level of indicated genes by RT-qPCR. (C) The stable cells overexpressing

p22 or wild-type (WT) IBRS-2 cells were lysed for western blotting analysis using the indicated antibodies. (D) HEK293T cells transfected with pFlag-KP177R was treated with IFN-γ. The cells were subjected to isolation of cellular total RNAs followed by quantification of transcription level of indicated genes by RT-qPCR. (E) HEK293T cells transfected with pFlag-KP177R was treated with 20 ng/mL IFN-γ for the indicated hours. Western blotting analysis was performed using the indicated antibodies.
(TIF)

**S3 Fig. The ASFV p22 degrades IFNAR1.** (A) HEK293T cells were cotransfected with the plasmids expressing HA-IFNAR1, -IFNAR2, -JAK1, -TYK2, -STAT1, -STAT2, or -IRF9, and Flag-p22 then lysed and analyzed using co-IP with anti-Flag MAb, followed by western blotting analysis using the indicated antibodies. (B) HEK293T cells were cotransfected with different amounts of the p22-expressing plasmid with pHA-β-actin, and pHA-IFNAR1 or pHA-IFNAR2, followed by western blotting analysis using the indicated antibodies at 24 hpt. The densitometric analysis of the protein expression levels was performed using the ImageJ software. (C) HEK293T cells were transfected with different amounts of the p22-expressing plasmid, and then the total RNA was extracted and analyzed using RT-qPCR at 24 hpt.
(TIF)

**S4 Fig. The ASFV p22 degrades IFNAR1 by inducing autophagy.** (A and B) HEK293T cells were cotransfected with different amounts of the p22-expressing plasmid with pHA-IFNAR1 and pHA-β-actin. Then, the cells were treated with DMSO, SBI, or 3-MA at 24 hpt, followed by western blotting analysis using the indicated antibodies. (C) HEK293T cells were transfected with siRNAs for 24 hours, followed by the cells were cotransfected with the plasmids pMyc-IFNAR1, pHA-β-actin, and pFlag-KP177R or pRK (Vec) and then western blotting analysis using the indicated antibodies. The densitometric analysis of the protein expression levels was performed using the ImageJ software.
(TIF)

**S5 Fig. TAX1BP1 is involved in the p22-induced degradation of IFNAR1.** (A) HEK293T cells were cotransfected with the plasmids expressing HA-ULK1, -ATG13, or -beclin-1, and Flag-p22, and then lysed for co-IP using anti-Flag MAb, followed by western blotting analysis using the indicated antibodies. (B) HEK293T cells were transfected with different amounts of the HA-TAX1BP1-expressing plasmid with pMyc-IFNAR1, pHA-β-actin, and pFlag-KP177R, followed by western blotting analysis using the indicated antibodies at 24 hpt. (C) The TAX1BP1-knockout or wild-type (WT) HEK293T cells were lysed for western blotting analysis using the indicated antibodies. (D) The *TAX1BP1*-knockout or WT HEK293T cells were transfected with different amounts of the Flag-p22-expressing plasmid at 24 hpt, followed by western blotting analysis using the indicated antibodies. The densitometric analysis of the protein expression levels was performed using the ImageJ software.
(TIF)

**S6 Fig. The transmembrane domain of the ASFV p22 interacting with IFNAR1 and TAX1BP1.** (A and B) HEK293T cells were cotransfected with the plasmids expressing Myc-IFNAR1 or HA-TAX1BP1 and Flag-p22 or the plasmids expressing the truncated Flag-p22 mutants, and then lysed for co-IP assay using anti-Flag MAb, followed by western blotting analysis using the indicated antibodies. (C) HEK293T cells were cotransfected with the plasmids expressing Flag-p22 and Myc-IFNAR1 or the plasmids expressing the truncated Myc-IFNAR1 mutants, and then lysed for co-IP assay using anti-Flag MAb, followed by western blotting analysis using the indicated antibodies. (D) HEK293T cells were cotransfected with the plasmids expressing Flag-p22 and HA-TAX1BP1 or the plasmids expressing the truncated HA-TAX1BP1 mutants and then lysed for co-IP assay using anti-Flag MAb, followed by western blotting analysis using the indicated antibodies.
(TIF)

**S1 Table. Primers used in this study.**
(DOCX)

**S2 Table. Primers used for constructing the transfer vector pOK12-p72-EGFP-ΔKP177R.**
(DOCX)

## Acknowledgments

The authors are grateful to Prof. Hong-Bing Shu (Wuhan University, China) for providing the HEK293T cells and the plasmids pNF-κB-Fluc, pSTAT1-Fluc, pRL-TK, psPAX2, and pMD2.0G, and the pLOV vector. We thank the Animal Biosafety Level 3 (ABSL-3) facilities in HVRI of CAAS for helpful experimental facilities and analysis support.

## Author contributions

**Conceptualization:** Hua-Ji Qiu, Gai-Ping Zhang, Su Li, Wen-Rui He.

**Data curation:** Haojie Ren, Yanjin Wang, Lian-Feng Li, Wen-Rui He.

**Formal analysis:** Haojie Ren, Yanjin Wang, Lian-Feng Li, Lan-Fang Shi, Yu-He Ma, Jun-Hao Fan, Wen-Rui He.

**Funding acquisition:** Gai-Ping Zhang, Wen-Rui He.

**Investigation:** Haojie Ren, Su Li, Wen-Rui He.

**Methodology:** Haojie Ren, Su Li, Wen-Rui He.

**Project administration:** Gai-Ping Zhang, Wen-Rui He.

**Resources:** Wen-Rui He.

**Software:** Yanjin Wang.

**Supervision:** Hua-Ji Qiu, Gai-Ping Zhang, Su Li, Wen-Rui He.

**Validation:** Haojie Ren, Yanjin Wang, Lian-Feng Li, Lan-Fang Shi, Xiao-Ya Pan, Han-Cheng Shao, Yuhang Zhang, Shichong Han, Bo Wan, Su Li.

**Visualization:** Haojie Ren, Yanjin Wang, Lian-Feng Li, Lan-Fang Shi, Xiao-Ya Pan, Han-Cheng Shao, Yuhang Zhang, Shichong Han, Bo Wan, Su Li.

**Writing – original draft:** Haojie Ren, Wen-Rui He.

**Writing – review & editing:** Haojie Ren, Hua-Ji Qiu, Gai-Ping Zhang, Su Li, Wen-Rui He.

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
