## [Decision Letter · Decision Letter 0]

The African swine fever virus p22 inhibits the JAK-STAT signaling pathway by promoting the TAX1BP1-mediated degradation of the type I interferon receptor

PLOS Pathogens

Dear Dr. He,

Thank you for submitting your manuscript to PLOS Pathogens. After careful consideration, we feel that it has merit but does not fully meet PLOS Pathogens's publication criteria as it currently stands. Therefore, we invite you to submit a revised version of the manuscript that addresses the points raised during the review process.

Please submit your revised manuscript within 60 days Jun 02 2025 11:59PM. If you will need more time than this to complete your revisions, please reply to this message or contact the journal office at plospathogens@plos.org. Please include the following items when submitting your revised manuscript:

We look forward to receiving your revised manuscript.

Kind regards,

Pinghui Feng

Academic Editor

PLOS Pathogens

Robert Kalejta

Section Editor

PLOS Pathogens

Editor-in-Chief

PLOS Pathogens

orcid.org/0000-0003-2946-9497

Editor-in-Chief

PLOS Pathogens

orcid.org/0000-0002-7699-2064

**Journal Requirements:**

At this stage, the following Authors/Authors require contributions: Lan-Fang Shi. Please ensure that the full contributions of each author are acknowledged in the "Add/Edit/Remove Authors" section of our submission form.

2) Please include an Ethics Statement for your study. Please ensure to incorporate:

i) The approval number(s), or a statement that approval was granted by the named board(s).

3) Some material included in your submission may be copyrighted. According to PLOSu2019s copyright policy, authors who use figures or other material (e.g., graphics, clipart, maps) from another author or copyright holder must demonstrate or obtain permission to publish this material under the Creative Commons Attribution 4.0 International (CC BY 4.0) License used by PLOS journals. Please closely review the details of PLOSu2019s copyright requirements here: PLOS Licenses and Copyright. If you need to request permissions from a copyright holder, you may use PLOS's Copyright Content Permission form.

Potential Copyright Issues:

i) Figure 9. Please confirm whether you drew the images / clip-art within the figure panels by hand. If you did not draw the images, please provide (a) a link to the source of the images or icons and their license / terms of use; or (b) written permission from the copyright holder to publish the images or icons under our CC BY 4.0 license. Alternatively, you may replace the images with open source alternatives. See these open source resources you may use to replace images / clip-art:

4) Please amend your detailed Financial Disclosure statement. This is published with the article. It must therefore be completed in full sentences and contain the exact wording you wish to be published.

**Reviewers' Comments:**

Reviewer's Responses to Questions

**Part I - Summary**

Reviewer #1: In this study, the authors described that ASFV internal envelope membrane protein P22 degrades IFNAR1 through autophagy pathway by relying on the selective autophagy receptor TAX1BP1, and inhibits the JAK-STAT signaling pathway. These findings clarify the biological functions of p22 in ASFV replication and uncover a novel antagonistic mechanism of the virus by autophagic degradation of IFNAR1. In general, this study is well designed, clearly presented, and provides compelling insights. However, several concerns should be addressed to improve the quality of the paper.

Reviewer #2: In the manuscript by Ren et al. (PPATHOGENS-D-25-00461), the authors identified the internal envelope membrane protein (p22) of ASFV as a negative regulator of JAK-STAT signaling pathway and conducted a series of experiments to show its role on IFNAR1 stability. The authors demonstrated that p22 interacted and promoted autophagic degradation of IFNAR1 by recruiting autophagy receptor TAX1BP1 to IFNAR1. Besides, they also demonstrated that ASFV-dKP177R virus failed to inhibit IFN-induced ISG expression compared to ASFV-WT infection, suggesting critical role of ASFV structural protein p22 to evade host immune response. Together this work convincingly showed that p22 regulates JAK-STAT signaling through autophagic degradation of IFNAR1.

Reviewer #3: Large knowledge gaps regarding the biological characteristics of African swine fever virus (ASFV) structural proteins have severely hindered the development of vaccines against African swine fever (ASF). In this study, Ren et al. investigated the functional role of viral structural protein p22 involved in the immune evasion of ASFV, and revealed that p22 inhibited the JAK-STAT signaling by promoting the TAX1BP1-mediated degradation of the type I interferon receptor. This study provides convincing evidence that the selective autophagy receptor TAX1BP1 is involved in interaction with p22 and IFNAR1 through biochemical and cell-based assays. Furthermore, the p22 was shown to promote the association of the TAX1BP1 with the IFNAR1 and facilitate the autophagic degradation of IFNAR1 to inhibit innate immunity. These findings are of significant interest to researchers in the fields of autophagy and virology. However, the following comments should be addressed to further refine the manuscript.

**Part II – Major Issues: Key Experiments Required for Acceptance**

Reviewer #1: Major concerns:

1. Fig 1C showed that P22 was at a low expression level at 24 hpi, while the RNA-seq data authors compared the difference between ASFV-ΔKP177R and ASFV-WT at 12 and 18 hpi, was the conclusion reached by the authors due to P22?

2. The effects of P22 on the expression of STAT1, STAT2, P-STAT1 and P-STAT2 were detected (Fig 2E, Fig 3C, Fig 8D). However, the expression of these proteins in the control group should be consistent, the authors should check these WBs.

3. In Fig 3D and Fig 8C, the fold-change of STAT1, IFIT1, and RSAD2 after treatment by IFN-β in IBRS-2 cells was not consistent, the stimulation difference should be described.

4. The LC3 lipidation experiments presented in Figure 5C and 5E should be repeated to enhance clarity, and the analyses of protein intensity should be included in the figure.

5. In Fig 5D, it showed that P22 promoted the co-localization of TAX1BP1 and LC3, and there were three colors in the figure. How did the author perform Pearson's R vale calculation should be described.

6. P22 induces autophagy in cells, why LC3 aggregates into the nucleus after addition of P22?

7. In Fig 5B, the expression of IFNAR1 was particularly up-regulated in the CQ treatment group?

8. Fig.8 The author wants to prove that the KP177R-TMD domain is its key functional domain. Why KP177R-ΔECD (including ICD and TMD) is chosen instead of KP177R-ΔTMD for the experiment?

9. A previous study demonstrated that TAX1BP1 was downregulated and cleaved by Coxsackievirus B3 (PMID: 34149637). The temporal expression profile of endogenous TAX1BP1 following ASFV infection should be systematically examined. Furthermore, this study should also rule out the potential effects of ASFV proteases on TAX1BP1 expression.

Reviewer #2: 1. The authors performed RNA-seq in PAM cells infected with ASFV-WT and ASFV-dKR177R virus and found that JAK-STAT signaling pathway is enriched by comparison of WT and dKR177R virus infection. The authors performed qPCR to validate the negative role of p22 in JAK-STAT signaling pathway, while the ISGs validated such as IFIT1�STAT1, RSAD2 is not the most upregulated genes from RNA-seq. Again, the JAK-STAT is not the top enriched pathways from the KEGG analysis. Validation of the top enriched pathways and genes are required to justify the functional significance of p22 depletion on ASFV induced immune response.

2. In their previous study, the authors demonstrated that IFN-gamma signaling is critical during ASFV infection in PAM cells. Does p22 of ASFV play any role in IFN-gamma triggered signaling pathway?

Reviewer #3: 1. The subcellular localization of p22 was analyzed in the cells with ectopic expression, and it is essential to examine the subcellular localization of p22 in ASFV-infected cells.

2. The authors should discuss which proteins, among pS273R or pMGF360-9L, pH240R, pMGF505-7R, and pF778R, primarily regulate the JAK-STAT signaling.

**Part III – Minor Issues: Editorial and Data Presentation Modifications**

Reviewer #1: Minor issues:

1. Introduction section lacks detailing knowledge on ASFV and autophagy, as well as related studies that explore the interaction between individual ASFV proteins and autophagy (PMID: 37566637, PMID: 37442088, PMID: 33830435, PMID: 33830435).

2. KP177R has previously been demonstrated to be non-essential (PMID: 34073222), which should also be included in the Introduction rather than the Discussion section.

3. If the KP177R deletion virus induces the upregulation of ISGs, could this imply that the virus is also inducing IFN? Alternatively, might the observed differences be due to the presence of type I IFNs in the supernatant of the KP177R-deleted virus preparations, which are absent in the wild-type virus?

4. TAX1BP1-ΔUBD already exists in Fig 7C, but its schematic diagram is shown in Fig 8A.

5. In Fig S5A and B, the expression of KP177R-ΔECD in input group was high, but it was low in IP group?

6. The format writing of the expression of P22 (KP177R) should be consistent throughout the text.

Reviewer #2: 1. The plasmids used in this study should clarify the sources. For example, pSTAT1-Fluc, it's originally from commercial resources or generated in the lab.

2. The author demonstrated that p22 reduced the phosphorylation levels of STAT1 and STAT2. Please clarify the phosphorylation types and residues of STATs, such as tyrosine phosphorylation (pY701), serine phosphorylation (PS727) of STAT1 or other type of phosphorylation antibody used in this study.

3. In line 207, STATs are transcription factors, not adaptor proteins.

4. In line 240, two “that” in the text.

Reviewer #3: 1. The IBRS-2 cell lines stably expressing p22 and TAX1BP1 knockdown cell lines were constructed in this study, it is essential to provide evidence in the Supporting Information confirming the successful establishment and validation of these stable cell lines.

2. Fig. 1F, A more detailed description of the assay and result is required to enhance the clarity.

3. The manuscript fails to adequately introduce the broader literature and understanding of virus interaction with autophagy. Several studies have shown that RNA viruses, including enteroviruses (doi�10.1080/15548627.2022.2153572), Coronavirus (doi�10.1080/15548627.2020.1817280) , Influenza A viruses (doi: 10.1080/15548627.2022.2162798), Picornavirus (doi�10.1080/15548627.2024.2350270), subvert the host autophagy pathway at multiple stages. Furthermore, there exist a substantial amount of literature on viral manipulation of selective autophagy receptors, including p62, T6BP, NDP52, and NBR1, which have not been discussed in this context.

4. Fig. 8C: There is a notable discrepancy between Figs. 8B and 8C regarding the JAK-STAT signaling of the dECD mutant. In Fig. 8B, the dECD mutant exhibited significantly higher inhibition of the signaling compared to the FL (full length) version, however, when examining their effects on ISGs expression, no significant difference is observed between the FL and the dECD mutant. The authors should provide an explanation for this observation in the Discussion section.

PLOS authors have the option to publish the peer review history of their article (what does this mean? ). If published, this will include your full peer review and any attached files.

**Do you want your identity to be public for this peer review?** For information about this choice, including consent withdrawal, please see our Privacy Policy .

Reviewer #1: No

Reviewer #2: No

Reviewer #3: No

**Figure resubmission:**

**Reproducibility:**



---

## [Decision Letter · Decision Letter 1]

Dear Dr. He,

We are pleased to inform you that your manuscript 'The African swine fever virus p22 inhibits the JAK-STAT signaling pathway by promoting the TAX1BP1-mediated degradation of the type I interferon receptor' has been provisionally accepted for publication in PLOS Pathogens.

Best regards,

Pinghui Feng

Academic Editor

PLOS Pathogens

Robert Kalejta

Section Editor

PLOS Pathogens

Sumita Bhaduri-McIntosh

Editor-in-Chief

PLOS Pathogens

orcid.org/0000-0003-2946-9497

Michael Malim

Editor-in-Chief

PLOS Pathogens

orcid.org/0000-0002-7699-2064

Reviewer Comments (if any, and for reference):

Reviewer's Responses to Questions

**Part I - Summary**

Reviewer #1: The authors have addressed all the concerns raised. These findings clarify a novel biological

function of p22 in ASFV replication, which is significant for understanding ASFV pathogenesis.

Reviewer #2: The authors have performed substantial experiments and discussion to address the reviewers' concern, which significantly improved the manuscript.

Reviewer #3: None

**Part II – Major Issues: Key Experiments Required for Acceptance**

Reviewer #1: All the concerns raised have been thoroughly addressed.

Reviewer #2: I thank the authors efforts for performing additional experiments to improve the manuscript. I have no additional comments for this manuscript.

Reviewer #3: none

**Part III – Minor Issues: Editorial and Data Presentation Modifications**

Reviewer #1: In the main text, the IFN-β protein should not be written in italics.

Reviewer #2: (No Response)

Reviewer #3: none

PLOS authors have the option to publish the peer review history of their article (what does this mean? ). If published, this will include your full peer review and any attached files.

**Do you want your identity to be public for this peer review?** For information about this choice, including consent withdrawal, please see our Privacy Policy .

Reviewer #1: No

Reviewer #2: No

Reviewer #3: No

---

## [Editor Report · Acceptance letter]

Dear Ms. He,

We are delighted to inform you that your manuscript, "The African swine fever virus p22 inhibits the JAK-STAT signaling pathway by promoting the TAX1BP1-mediated degradation of the type I interferon receptor," has been formally accepted for publication in PLOS Pathogens.

Best regards,

Sumita Bhaduri-McIntosh

Editor-in-Chief

PLOS Pathogens

orcid.org/0000-0003-2946-9497

Michael Malim

Editor-in-Chief

PLOS Pathogens

orcid.org/0000-0002-7699-2064